

# On the Configuration and Initialization of a Large Scale Hydrological Land Surface Model to Represent Permafrost

Mohamed E. Elshamy[1*], Daniel Princz[2], Gonzalo Sapriza-Azuri[3], Al Pietroniro[1,2], Howard S. Wheater[1], and
Saman Razavi[1]
[1] Global Institute for Water Security, University of Saskatchewan, 11 Innovation Blvd, Saskatoon, SK,
Canada S7N 3H5
[2] Environment and Climate Change Canada, 11 Innovation Blvd, Saskatoon, SK, Canada S7N 3H5
[3] Departamento del Agua, Centro Universitario Regional Litoral Norte, Universidad de la República, Salto,
Uruguay
*Corresponding author: mohamed.elshamy@usask.ca

## Abstract

Permafrost is an important feature of cold regions hydrology, particularly in basins such as the Mackenzie
River Basin (MRB), and needs to be properly represented in hydrological and land surface models (H-LSMs)
built into existing Earth System models (ESM), especially under the unprecedented climate warming
trends that have been observed. Higher rates of warming have been reported in high latitudes compared
to the global average resulting in permafrost thaw with wide-ranging implications for hydrology and
feedbacks to climate. The current generation of H-LSMs is being improved to simulate permafrost
dynamics by allowing deep soil profiles and incorporating organic soils explicitly. Deeper soil profiles have
larger hydraulic and thermal memories that require more effort to initialize. This study aims to devise a
robust, yet computationally efficient, initialization and parameterization approach applicable to regions
where data are scarce and simulations typically require large computational resources. The study further
demonstrates an upscaling approach to inform large-scale ESM simulations based on the insights gained
by modelling at small scales. We used permafrost observations from three sites along the Mackenzie River
Valley spanning different permafrost classes to test the validity of the approach. Results show generally
good performance in reproducing present-climate permafrost properties at the three sites. The results
also emphasize the sensitivity of the simulations to the soil layering scheme used, the depth to bedrock
and the organic soil properties.

## Keywords

Hydrological Land Surface Models, Permafrost, Initialization, Organic Soils, Mackenzie River Basin





## 1. Introduction

Earth system models (ESMs) are widely used to project climate change and they show a current global
warming trend that is expected to continue during the 21$^{st}$ century and beyond (IPCC, 2014). Higher rates
of warming have been observed in high latitudes compared to the global average (DeBeer et al., 2016;
McBean et al., 2005) resulting in permafrost thaw with implications for soil moisture, hydraulic
connectivity, streamflow seasonality, land subsidence, and vegetation (Walvoord and Kurylyk, 2016).
Recent analyses provided by Environment and Climate Change Canada (Zhang et al., 2018) have shown
that Canada's far north has already seen an increase in temperature of double the global average, with
some portion of the Mackenzie basin already heating up by 4°C. Subsequent impacts on water resources
in the region however, are not so clear.  Recent analysis of trends in Arctic freshwater inputs (Durocher
et al., 2019) highlights that Eurasian rivers show a significant annual discharge increase during 1975-2015
period while in North American rivers; only rivers flowing into the Hudson Bay region in Canada show a
significant annual discharge change during that same period. Those rivers in Canada flowing directly into
the Arctic, of which the Mackenzie River provides the majority of flow, show very little change.
Deep uncertainty in hydrological response to a changing climate is resulting from poor understanding and
characterization of cold-regions processes in ESMs. Despite advances in cold-regions process
understanding and modelling at the local scale (e.g. Pomeroy et al., 2007), their upscaling and systematic
evaluation over large domains remain rather elusive. This is largely due to lack of observational data, the
local nature of these phenomena and the complexity of cold-region systems. Hydrological response and
land-surface feedbacks in cold-regions are generally complex and depend on a multitude of several inter-
related factors including changes to precipitation intensity, timing, and phase as well as soil composition
and hydraulic and thermal properties. As permafrost underlies about one quarter of the exposed land in
the Northern hemisphere (Zhang et al., 2008), it is imperative to study and accurately model its behaviour
under current and future climate conditions. Knowledge of permafrost conditions (temperature, active
layer thickness, and ground ice conditions) and their spatial and temporal variations is critical for planning
of development in Northern Canada (Smith et al., 2007) and other Arctic environments.
There has been extensive regional and global modelling efforts which involve cold-region processes
including permafrost (Riseborough et al., 2008; Walvoord and Kurylyk, 2016). These studies, however,
typically focused on and modeled only a shallow soil profile in the order of a few meters. For example,
the Canadian Land Surface Scheme (CLASS) typically uses 4.1m (Verseghy, 2012) and the Joint UK Land
Environment Simulator (JULES) standard configuration is only 3.0m (Best et al., 2011). These are too




shallow to represent permafrost properly and could result in misleading projections. For example,
Lawrence and Slater (2005) used a 3.43m soil column to project the impacts of climate change on near-
surface permafrost degradation in the Northern hemisphere using the Community Climate System Model
(CCSM3), which lead to overestimation of climate change impacts and raised considerable criticism (e.g.
Burn and Nelson, 2006).  It eventually lead to further development of the Community Land Model (CLM),
the land surface scheme of the CCSM, to include deeper soil profiles (e.g. Swenson et al., 2012).
Recognizing this issue, more recent studies have indicated the need to have a deeper soil column (20-25m
at least) in land surface models (run stand-alone or embedded within ESMs) than previously used, to
properly capture changes in freeze and thaw cycles and active layer depth dynamics (Lawrence et al.,
2012; Romanovsky and Osterkamp, 1995; Sapriza-Azuri et al., 2018).
However, a deeper soil column implies larger soil hydraulic and, more importantly, thermal memory that
requires proper initialization to be able to capture the evolution of past, current and future changes. Initial
conditions are established by either spinning up the model for many annual cycles (or multi-year historical
cycles, sometimes de-trended) to reach some steady state or by running it for a long transient simulation
for 100s of years or both (spinning to stabilization followed by a long transient simulation). Lawrence et
al. (2008) spun up CLM3.5 for 400 cycles with year 1900 data for deep soil profiles (50-125m) to assess
the sensitivity of model projections to soil column depth and organic soil representation. Park et al. (2013)
used 21 cycles of the first 20 years of the climate record they used (1948-2006) to initialize their CHANGE
land surface model to study differences in active layer thickness between Eurasian and North American
watersheds. However,  Ednie et al. (2008) implied from borehole observations in the Mackenzie Valley
that present day permafrost is in disequilibrium with current climate, and therefore, it is unlikely that we
can establish a reasonable representation of current ground thermal conditions by employing present or
20[th] century climate conditions to start the simulations. Nevertheless, their analysis of paleo-climatic
records (Szeicz and MacDonald, 1995) of summer temperature at Fort Simpson, dating back to the early
1700s, shows that a negative (cooling) trend prevailed till mid 1800s followed by a positive (warming)
trend till present and they "assumed" a quasi-equilibrium period prior to 1720. Using that assumption,
they used an equilibrium thermal model called T-TOP to establish the initial conditions of 1721 and then
the temperature trends thereafter to carry out a transient simulation till 2000 using the T-ONE thermal
model. Those thermal models use air temperature as their main input while land surface models (as used
here and described below) require a suite of meteorological inputs. Sapriza-Azuri et al. (2018) used tree-
ring data from Szeicz and Macdonald (1995) to construct climate records for all variables required by
CLASS at Norman Wells in the Mackenzie Valley since 1638 to initialize the soil profile of their model.




While useful, such proxy records are not easily available at most sites. Additionally, re-constructing several
climatic variables from summer temperature introduces significant uncertainties that need to be
assessed.  Thus, there is a need to formulate a more generic way to define the initial conditions of soil
profiles across large domains.
Additionally, concerns are not only about the depth of the whole profile. The definition of the layer
thicknesses requires due attention. Land surface models that utilize deep soil profiles exponentially
increase the layer thicknesses to reach the total depth using a reasonably tractable number of layers (15-
20). For example, CLM 4.5 (Oleson et al., 2013) used 15 layers to reach a depth of 42.1m for the soil
column. Sapriza-Azuri et al. (2018) used 20 layers to reach a depth of 71.6m in their experiments using
MESH/CLASS. Park et al. (2013) had a 15-layer soil column with exponentially increasing depth to reach a
total depth of 30.5m in the CHANGE land surface model. However, the first version of CHANGE had only
11m soil column depth (Park et al., 2011).
The importance of insulation from the snow cover on the ground and/or organic matter in the upper soil
layers is key to the quality of ALD simulation results (Lawrence et al., 2008; Park et al., 2013). Organic soils
have large heat and moisture capacities that, depending on their depth and composition, moderate the
effects of the atmosphere on the deeper permafrost layers and work all year round. Snow cover, in
contrast, varies seasonally and inter-annually and can thus induce large variations to the ALD, especially
in the absence of organic matter (Park et al., 2011). Climate change impacts on precipitation intensity,
timing, and phase are translated to permafrost impacts via changing the snow cover period, spatial extent,
and depth. Therefore, it is critical to the simulation of ALD that the model includes organic soils and has
adequate representation of snow accumulation (including sublimation and transport) and melt processes.

## 2. Objectives

The main objective of the present study is to devise an approach to configure and initialize the soil profile
of a land surface model to account for permafrost in large-scale applications. The elements of this strategy
include:
-  Defining how deep should the soil profile be, to allow proper simulation of the ALD dynamics for
current and future climate.
-  Determining the appropriate vertical discretization to give enough accuracy in determining the
ALD while optimizing computational resources for large-scale applications. This also includes



configuring the organic layers (how many, which properties, etc.) and the depth to bedrock (see
description below).
-    Determining how to initialize the deep soil profile, whether cycling a single year or multiple years
and finding the appropriate number of cycles. In addition to studying the sensitivity of
performance to the selected year(s) for spinning.
This study is part of a larger study that aims to develop a large-scale hydrological model for the Mackenzie
River Basin (MRB) (Figure 1) using the MESH (Modélisation Environmentale Communautaire - Surface and
Hydrology) framework and validate the model in order to use it to study climate and land use/cover
change impacts on various aspects of its hydrology. Permafrost underlies 70-80% of the MRB and thus it
exerts considerable control on its hydrology, especially in a warming climate. The next section describes
the model briefly and the datasets and methods used in the study. Section 4 displays the results of the
analyses that are discussed in Section 5 with some concluding remarks.
**Possible position for Figure 1**

## 3. Models, Methods, and Datasets

### 3.1 The MESH Modelling Framework

MESH is a semi-distributed hydrological-land surface model (H-LSM) coupled with streamflow routing
(Pietroniro et al., 2007). It has been widely used in Canada to study the Great Lakes Basin (Haghnegahdar
et al., 2015) and the Saskatchewan River Basin (Yassin et al., 2017) amongst others. Several applications
to basins outside Canada are underway (e.g. Arboleda-Obando, 2018; Bahremand et al., 2018). The MESH
framework allows coupling of a land surface model, either CLASS (Verseghy, 2012) or SVS (Husain et al.,
2016) that models the vertical processes of heat and moisture flux transfers between the land surface and
the atmosphere, with a horizontal routing component (WATROUTE) taken from the distributed
hydrological model WATFLOOD (Kouwen, 1988). Unlike most land surface models, the vertical column has
a slope that allows for lateral transfer of overland and interflow (Soulis et al., 2000) to an assumed stream
within each grid cell of the model. MESH uses a regular latitude-longitude grid and represents subgrid
heterogeneity using the grouped response unit (GRU) approach (Kouwen et al., 1993) which makes it
semi-distributed. In the GRU approach, different land covers within a grid cell do not have specific
locations and do not interact explicitly, making it easier for parameterization. While, Land cover classes
are typically used to define a GRU, other factors can be included in the definition such as soil type, slope,
aspect, etc. A tile, which is the smallest computational element, is defined by a specific GRU in a given grid



cell. MESH has been under continuous development; its new features include improved representation of
baseflow (Luo et al., 2012), controlled reservoirs (Yassin et al., 2019) as well as permafrost (this paper).
More details about MESH history and developments are provided in a companion paper (Davison et al.,
in preparation). For this study, we use CLASS as the underlying land surface model within MESH.
Underground, CLASS couples the moisture and energy balances for a pre-specified number of soil layers
of pre-specified thicknesses. Each soil layer, thus, has a diagnosed temperature and both liquid and frozen
moisture contents down to the soil permeable depth or the "depth to bedrock – SDEP" below which there
is no moisture and the thermal properties of the soil are assumed as those of bedrock material
(sandstone). MESH is usually run at 30min time steps and thus from the MESH-simulated continuous
temperature profiles, one can determine several permafrost related aspects that are used in the analyses
such as (see Figure 2):
-    Temperature envelopes at daily, monthly and annual time steps. Temperature envelopes are

defined by the maximum and minimum simulated temperature for each layer over the specified

time period.

-    Active layer thickness (or depth – ALD) defined as the maximum depth of the zero isotherm over

the year taken from the annual temperature envelopes by linear interpolation between layers

bracketing the zero value (freezing point depression is not considered). It has to be connected to

the surface, thus we use a thaw, rather than freeze, criterion, which is compatible with the

available measurements.

-    Daily progression of the ALD, which can be used to visualize the thaw and freeze fronts and

determine the dates of thaw and freeze-up. These are calculated in a similar way to the annual

ALD but using the daily envelopes.

-    The no (or zero) oscillation depth (ZOD) where the annual temperature envelopes meet to within

0.1° (or other given accuracy threshold). In some literature, this depth is termed the zero

amplitude depth (ZA).

**Possible position for see Figure 2**

Permafrost is usually defined as ground remaining frozen for at least two years but for modelling purposes
and to validate against annual ground temperature envelope and ALD data, a one-year cycle is adopted.
This is common amongst the climate and land surface modelling community (e.g. Park et al., 2013).
MESH/CLASS used to output temperature profiles; the code has been amended to calculate the additional
outputs detailed above for each tile as well as the grid average allowing spatial and temporal mapping of



permafrost characteristics. A CLASS typical configuration consists of 3 soil layers of 0.1, 0.25, and 3.75m
thickness but in 2006, it was extended to accommodate as many layers as needed (Verseghy, 2012).
However, this was hard-coded within CLASS until it became configurable using an external file only within
the MESH framework. The configuration file used to provide soil parameters (texture and initial
temperature and moisture conditions) for each GRU for the top three layers and the model assumed the
third layer values to apply to any additional layers below till bedrock. The code has been modified to
enable specifying these parameters for as many layers as needed and was extended to allow a spatially
variable specification (i.e. by grid) of these parameters as well as by GRU. However, the number and
thickness of soil layers are still fixed for the whole domain.
Organic soils are modelled in CLASS by deactivating mineral soils using a special flag to allow a soil layer
to either be Fibric, Hemic, or Sapric after Letts et al. (2000). Each type has a different degree of
decomposition leading to different physical, hydraulic and thermal properties as specified in Verseghy
(2012). Usually, a soil layer is assumed to be fully organic if the organic content is 30% or more (Soil
Classification Working Group, 1998). Organic soils were mapped from the Soil Landscapes of Canada (SLC)
v2.2 (Centre for Land and Biological Resources Research, 1996) for the whole MRB (Figure 3). However,
this dataset does not provide information as to the depth of the organic layers or their configuration (i.e.
the thicknesses of Fibric, Hemic and Sapric layers). Therefore, different configurations have been tested
at the study sites based on available local information keeping in mind that these has to be carried back
to the MRB scale.
**Possible Position for Figure 3**

## 3.2 Study Sites and Data

The Mackenzie River Basin (MRB) extends between 102-140°W and 52-69°N (Figure 1). It drains an area
of about 1.775 Mkm$^2$ of Western and Northwestern Canada and covers parts of Saskatchewan, Alberta,
British Columbia provinces as well as the Yukon and the North West Territories. The average annual
discharge at the basin outlet to the Beaufort Sea exceeds 300 km$^3$, which is the fifth largest discharge to
the Arctic. Such a large discharge influences regional as well as global circulation patterns under the
current climate, and is expected to have implications for climate change. Figure 1 also shows the
permafrost extent and categories for the MRB taken from the Canadian Permafrost Map (Hegginbottom
et al., 1995). About 75% of the basin is underlain by permafrost that can be either continuous (in the far
North and the Western Mountains), discontinuous (to the south of the continuous region), or sporadic (in
the southern parts of the Liard and in the Hay sub-basin). It is important, while building the MRB model,





to properly represent permafrost, given the current trends of thawing and its vast impacts on landforms,
connectivity, and thus the hydrology of the basin. This is the focus of this paper, through detailed studies
conducted at three sites on a transect near the Mackenzie River going from the Sporadic permafrost zone
(Jean Marie River) to the Extensive Discontinuous zone (Norman Wells) and the Extensive Continuous
zone (Havikpak Creek) as shown Figure 1. The following sections give a closer look at each site, the data
available, and some of the previous work conducted, focusing on permafrost.
*3.2.1  Jean Marie River*
The Jean Marie River (JMR) is a tributary of the main Mackenzie River Basin (Figure 4) in the Northwest
Territories (NWT) province of Canada. Its mouth is located upstream of Fort Simpson where the Liard River
joins the main Mackenzie River. The gauged area up to the WSC station at the river intersection with
Highway 1 is about 1240 km$^2$. The basin is dominated by boreal (deciduous, coniferous and mixed) forest
on raised peat plateaux and bogs. The basin is located in the sporadic permafrost zone characterized with
warm permafrost (temperature > -1°C) that underlies some parts and does not exist in others with limited
(<10m) thickness (Smith and Burgess, 2002).
**Possible Position for Figure 4**
The nearest Environment and Climate Change Canada (ECCC) Weather station is located at Fort Simpson
to the North of the Basin. The Canadian Climate Normals (1981–2010, ECCC) at Fort Simpson indicates
that the mean annual temperature is -2.8°C with temperatures generally below freezing during October
to April while a maximum summer temperature of 17.4°C is reached in July. Mean annual precipitation is
about 388 mm/year, of which around 60% falls as rain while the rest is snowfall.
The streamflow at Water Survey of Canada (WSC) gauge 10FB005 has a good record for the period 1972-
2015. The basin is snow-melt dominated with flow peaks normally occurring in May/June with some years
having secondary summer peaks. The mean annual streamflow at the station over the period 1980-2015
is 5.5 m$^3$/s, while the highest recorded streamflow reached 211 m$^3$/s on July 3, 1988. Baseflow is usually
small but the river does not run completely dry in winter despite surface freezing.
The gauged part of the basin, modelled for this study, is covered by 14 grid cells of the MRB model grid
(0.125° x 0.125°) and can thus be hydrologically assessed in terms of the quality of the streamflow
simulations. However, this is not the main focus of this study. Parameters for the MESH model are taken
from calibrations of the adjacent Liard sub-basin (Elshamy et al., in preparation).



The basin and adjacent basins (e.g. Scotty Creek) have been subject to extensive studies as the warm, thin,
and sporadic permafrost underling the region has been rapidly degrading (Calmels et al., 2015; Quinton
et al., 2011). The region is vulnerable to permafrost thaw, which is changing the landscape of the region,
the vegetation, and wildlife habitat with significant implications for First Nations livelihoods and access to
their cultural resources. Collapse of forested peat plateaux into wetland areas has been reported by
several researchers (e.g. Calmels et al., 2015; Quinton and Baltzer, 2013)
Several permafrost-monitoring sites have been established in and around the basin mostly as part of the
Norman Wells to Zama pipeline monitoring program launched by the Government of Canada and Enbridge
Pipeline Inc. in 1984-1985 to investigate the impact of the pipeline on the permafrost and terrain
conditions  (Smith et al., 2004). The details of those sites are given in Table 1 while Figure 4 shows their
locations. We focus on sites 85-12A and 85-12B as representative of the basin. We use Cables T4 at each
site as they are the least affected by the pipeline, being out of its right of way (at least 20m away). Site
85-12A has no permafrost while site 85-12B, in close proximity, has a thin (3-4m) permafrost layer with
ALD of about 1.5m as estimated from soil temperature envelopes over the period 1986-2000. All other
monitoring points on Figure 4 have no permafrost conditions since their records began in the 1980s and
1990s. The sites 85-12A & B have a ground moraine landform with open black spruce, ericaceous shrubs,
moss-lichen woodland on a peat plateau (Smith et al., 2004). It is challenging to model two different
conditions in such close proximity (within the same model grid cell and having the same vegetation). The
difference in permafrost conditions is possibly related to the thickness of the peat as shown in the
borehole logs (Smith et al., 2004). Borehole 85-12A-T4 has a little over 1m thick layer of peat while
borehole 85-12B-T4 has close to 5m peat providing more insulation that keeps the ground from thawing
during summer.
**Possible Position of Table 1**
*3.2.2  Bosworth Creek (Norman Wells)*
Bosworth Creek (BWC) is a small basin (126 km$^2$) on the Eastern/Northern Side of the Mackenzie River
(Figure 5) draining to the main Mackenzie river near Norman Wells. Permafrost monitoring activities
started in the region in 1984 with the construction of the Norman Wells to Zama buried oil pipeline as
mentioned above. The basin is dominated by boreal (deciduous, coniferous and mixed) forest. It is located
in the extensive discontinuous permafrost zone with relatively deep active layer (1-3 m) and relatively
thick (10-50m) permafrost (Smith and Burgess, 2002)





There is an ECCC weather station nearby at Norman Wells with complete temperature and precipitation
records from 1980. The Canadian Climate Normals (1981–2010, ECCC) at Norman Wells indicate that the
mean annual temperature is -5.1°C with temperatures generally below freezing during October to April
while the maximum summer temperature of 17.1°C is reached in July. Mean annual precipitation is about
294 mm/year, of which around 60% falls as rain while the rest is snowfall.
Similar to the Jean Marie River Basin, the streamflow is dominated by snowmelt with a peak in May and
a secondary summer peak in some years. WSC Gauge 10KA007 at the outlet of the basin near its
confluence with the Mackenzie River has a good record over the period 1980-2016 with a long gap from
1995-2008. The mean annual discharge over the available period of record is 0.67 $m^3$/s with peaks ranging
normally between 2.5 and 15 $m^3$/s. The highest daily flow on record reached about 20 $m^3$/s in May 1991.
There is a visible baseflow component for this basin. The basin covers portions of three grid cells of the
MRB grid (Figure 5) and therefore it is not expected to have adequate simulation for streamflow
comparisons.
**Possible Position of Figure 5**
The basin itself has not been the focus of previous hydrological studies, but there are several permafrost
studies of Norman Wells, being at the Northern end of the important pipeline. Sapriza-Azuri et al. (2018)
used cable T5 at the pump station site (84-1) to investigate the appropriate soil depth and initial conditions
for permafrost simulations, which is a pre-cursor for this current study. They recommend a soil depth of
a least 20m to ensure that the simulated ZOD is within the soil profile. However, they based their analysis
on cable T5, which is within the right of way of the pipeline and is likely to be affected by its
construction/operation.
There are several thermal monitoring sites within and close to the basin and the adjacent Canyon Creek
basin to its south East – Table 1. There are also a few thaw tubes but their records are short and
intermittent. We focus on the Norman Wells pump station site (84-1) and for this study we choose cable
T4 as it is more likely to reflect the natural permafrost conditions being out of the right of way of the
pipeline. It has a continuous record since 1985 (Smith et al., 2004; Duchesne, personal communication,

2017).

*3.2.3   Havikpak Creek*
Havikpak Creek (HPC) is a small arctic research basin (about 15 $km^2$ in area) located in the Eastern part of
the Mackenzie River basin delta, 2km north of the Inuvik Airport (68°18'15" N, 133°28'58" W) in the




Northwest Territories (NWT) (Figure 6). The basin is dominated by sparse taiga forest and shrubs, has a
cold sub-arctic climate and is underlain by thick permafrost (>300m). The basin is characterized by mild
slopes and has an elevation ranging between 60-240m (Krogh et al., 2017).

**Possible Position of Figure 6**

There is an ECCC weather station at nearby Inuvik airport with hourly temperature record from 1980 and
daily precipitation record from 1960. The Canadian Climate Normals (1981–2010, ECCC) at Inuvik indicates
that the mean annual temperature is -8.2°C with temperatures generally below freezing during October
to April while a maximum summer temperature of 14.1°C is reached in July. Mean annual precipitation is
about 241 mm/year; close to half of which is rainfall while the rest falls as snow.
The streamflow flow of the basin is dominated by snowmelt with no winter streamflow due to the lack of
groundwater contribution (deep permafrost), and some smaller summer events. The streamflow at the
outlet of the basin has been measured by ECCC WSC gauge 10LC017 since 1995. The mean annual
streamflow at the outlet is about 0.07 $m^3$/s with a maximum of 4.65 $m^3$/s reached in the summer of 2000.
The summer peak discharge varied greatly between 0.7 and 4.0 $m^3$/s over the period 1995-2017. However,
the basin covers portions of only two grid cells of the MRB grid (Figure 6) and therefore is not expected
to have adequate simulation for streamflow comparisons.
The basin has been subject to several hydrological studies, especially during the Mackenzie GEWEX Study
(MAGS). For example, Marsh et al. (2002) studied the water and energy fluxes from HPC for the important
1994/95 hydrological year. More recently, Krogh et al. (2017) modelled its hydrological and permafrost
conditions using the Cold Regional Hydrological Model (CRHM) (Pomeroy et al., 2007). They integrated a
ground freeze/thaw algorithm called XG (Changwei and Gough, 2013) within CRHM to simulate the active
layer thickness and the progression of the freeze/thaw front with time but they did not attempt to
simulate the temperature envelops or the depth/temperature of ZOD.
In terms of permafrost-related measurements, soil temperature envelopes are available from Inuvik
airport forest and bog sites 01TC02 and 01TC03 respectively. Ground temperatures are measured with
multi-sensor temperature cables installed in boreholes going down to 10m and 6.5m in depth at 01TC02
and 01TC03 respectively and both are equipped with data loggers (Smith et al., 2016). Temperature
sensors failed on the bog site (01TC03) in 2010 and the site was replaced by 12TC01 in the same
conditions.  In addition, there are three thaw tubes at Inuvik Upper Air station (90-TT-16) just to the west
of the basin,  at HPC (93-TT-02), and at the Inuvik Airport bog site (01-TT-03) measuring the active layer



depth and ground settlement (Smith et al., 2009). The land form and vegetation at Inuvik Airport forest
site (01TC02) is described as fluted till plain with open black spruce trees while the other site (01TC03) is
an open bog between ridges on the fluted till plain with scattered shrubs in an open bog. The HPC thaw
tube is located in a back spruce forest (Smith et al., 2009).

### 3.3 Soil Profile and Organic Soils

As mentioned earlier, Sapriza-Azuri et al. (2018) recommended a total soil column depth (D) of no less
than 20m to enable reliable simulation of permafrost dynamics considering the uncertainties involved
including parameter uncertainty. Their study is relevant because they used the same model used here
(MESH/CLASS). They studied several profiles, down to 71.6m depth. Recent applications of other H-LSMs
also considered deep soil column depths; e.g. CLM 4.5 used 42.1m (Oleson et al., 2013) and CHANGE (Park
et al., 2013) used 30.5m. After a few test trials with D = 20, 25, 30, 40, 50 and 100m at the different sites,
we found that the additional computation time when adding more layers to increase D is outweighed by
the reliability of the simulations. The reliability criterion used here is that the temperature envelopes meet
well within the soil column depth over simulation period (including spinning-up) such that the bottom
boundary condition is not disturbing the simulated temperature profiles/envelopes and ALD (Nicolsky et
al., 2007). ZOD (refer to Section 3.1) represents a relatively stable condition to assess that (Alexeev et al.,
2007). ZOD reached a maximum of 25m at one of the sites in a few years and thus the total depth was
increased to 50m in anticipation for possible changes in ZOD with warming. We show that this depth is
adequate at the three sites selected in the subsequent sections.
The CLASS thermal boundary condition at the bottom of the soil column is either no-flux (i.e. the gradient
of the temperature profile should be zero) or a constant geothermal flux. For this study, we considered
the no-flux condition, as data for the geothermal flux are not easy to find at the MRB scale. Nicolsky et al.
(2007) ignored the geothermal flux in their study over Alaska using CLM with an 80m soil column. Sapriza-
Azuri et al. (2018) showed that the difference in temperature at ZOD between the two cases is within the
error margin for geothermal temperature measurements for 60% of their simulations at Norman Wells.
The total soil column depth is only one factor in the configuration of the soil. The layering is as critical. In
the above-mentioned modelling studies, exponentially increasing soil layer thicknesses were used, aiming
to reach the required depth with a minimum number of layers. The exponential formulation creates more
layers near the surface, which allows the models to capture the strong soil moisture and temperature
gradients there and yet have a reasonable number of layers (15-20) to reduce the computational burden.
However, for most of the MRB, the observed ALD is in the range of 1-2m from the surface and the



exponential formulations increase layer thickness quickly after the first 0.5-1.0m, which reduces the
accuracy of the models, especially for transient simulations. Therefore, we adopted two layering schemes
that have more layers in the top 2m, and increased the layer thickness at lower depths, to 50m. The first
scheme has the first meter divided into 10 layers, the second meter divided into 5 layers and the total soil
column has 23 layers. The second scheme has soil thicknesses increasing more gradually to reach 51.24m
in 25 layers following a scaled power law. This latter scheme has an advantage that each layer is always
thicker than the one above it (except the second layer) which showed improvements in numerical stability
for both temperature and moisture calculations. The minimum soil layer thickness is taken as 10cm as
advised by Verseghy (2012) for numerical reasons. CLASS uses an explicit forward difference numerical
scheme to solve the energy and water budgets, which can have instabilities when layers have the same
thickness. Table 2 shows the soil layer thickness and centers (used for plotting temperature
profiles/envelopes) for both schemes.
**Possible Position of Table 2**
Finally, the discretization of organic soil is considered separately for each basin based on local information
together with the gridded SLC v2.2 at 0.125° resolution (Keshav et al., 2019a). The flexibility of the model
can be utilized for the selected basins when modelled separately but to take the information back to the
whole MRB, one has to rely on more general information that is available basin-wide. As discussed above,
CLASS (Verseghy, 2012) originally configured the first layer as fibric (type 1), the second as hemic (type 2)
and the rest as sapric (type 3) as soon as the organic soil flag is activated. We modified that to be
configurable such that one can have more than one fibric or hemic layer and switch off the organic soils
for the lower layers. Typically we use them in the same order as it reflects the natural decomposition
process (fibric at the surface, followed by hemic, then sapric) but with the introduction of configurable
layer depths, texture, and initial conditions, it is necessary to have organic layers configurable as well.
Fully organic soils are activated when the organic content is 30% or more (Soil Classification Working
Group, 1998).
For JMR, we tested configurations with about 0.6m organic soil (6 layers using SC1 and 5 under SC2) to
over 2m of organic soil. The soil is assumed to be uniform below the fully organic layers and the soil texture
is taken from the gridded SLC v2.2 mapping for the MRB mentioned above giving 15% SAND and 15% CLAY
and an organic content ranging between 48-59% (Figure 3). 4-7m peat depths have been reported in the
surrounding region (Quinton et al., 2011) and by borehole data of the specific permafrost monitoring sites
(Smith et al., 2004). Therefore, the organic content in the mineral layers below the fully organic layers is



set to 50% until bedrock. This is an exception for this basin which can be generalized for the MRB for high
organic content (e.g. > 50%) like this region. The organic configurations used are listed in Table 3. SDEP is
set to 7m based on gridding the Shangguan et al. (2017) dataset at the 0.125° resolution (Keshav et al.,
2019b). As mentioned in Section 3.1, SDEP marks the hydrologically active horizon below which the soil is
not permeable and its thermal properties are changed to those of bedrock material. This makes it an
important parameter and the sensitivity of the results to it is assessed by perturbing it within a range (5-
15m).
**Possible Position of Table 3**
For BWC, the organic map (Figure 3) indicated that organic matter ranges between 27-34%. We tested
configurations with 0.3 – 0.8m organic layers. A borehole log for 84-1-T4 site (Smith et al., 2004) shows a
thin organic silty layer at the top (close to 0.2-0.3m). Sand and clay content below the organic layers are
uniformly taken to be 24% and 24% respectively based on the gridded SLC v2.2 as above and the
remainder (52%) is assumed to be silt by CLASS. SDEP ranges between 5-12m. Thus, several values within
this range have been tested.
The organic content indicated by the gridded soil information at HPC is only 18%, which is lower than the
30% threshold to activate fully organic soils. However, Quinton and Marsh (1999) used a 0.5m thick
organic layer in their conceptual framework developed to characterise runoff generation in the nearby
Siksik creek. Krogh et al. (2017) adopted the same depth for their modelling study of HPC. Therefore, we
tested configurations with 0.3-0.8m fully organic layers. Below that, soil texture values are taken from the
gridded SLC v2.2 to be 24% Sand and 32% Clay. A mineral soil configuration with 18% organic matter for
the top few layers has been also tested (denoted "M-org"). SDEP ranges between 8-10m but values
ranging between 5-12m have been tested.

## 3.4 Land Cover Parameterization

As noted above, the model parameters for the three selected basins were pre-specified, given the specific
aims of this study. The setups use land cover, vegetation, and hydrology parameters from the MRB setup,
which is described in Elshamy et al. (in preparation). The land cover data are based on the CCRS 2005
dataset (Canada Centre for Remote Sensing (CCRS) et al., 2010) and the calibration differentiates between
the Eastern and Western sides of the basin using the Mackenzie River as a divide. HPC and BWC are on
the East side of the river while JMR is on the west side and therefore they have different parameters for
some GRU types (e.g. Needleleaf Forest). SDEP, soil texture information and initial conditions were taken



as described above and adjusted according to model evaluation versus permafrost related observations
(ALD, Temperature envelopes) with the aim to develop an initialization and configuration strategy that
can be implemented for the larger MRB model.
Special land covers within the MESH framework include inland water, which is parameterized such that it
remains saturated. Thus, drainage is prohibited from the bottom of the soil column and it is modelled
using flat CLASS (no slope) with a large hydraulic conductivity value. Ideally, water should have no
limitation on evaporation but being still treated as a porous media within the current version of CLASS,
the top layers are not always fully saturated. Additionally, it was initialized to have a positive bottom
temperature and therefore, it does not develop permafrost. Wetlands are treated in a similar way
(impeded drainage and no slope) but it has grassy vegetation and it takes the soil properties as described
above (Section 3.3). It remains close to saturation but, depending on location, can still be underlain by
permafrost. Taliks are easier to develop under wetlands this way.

## 3.5 Climate Forcing

MESH requires climate forcing data for seven climatic variables at a sub-daily time step. For this study we
used the WFDEI dataset that covers the period 1979-2016 at 3 hourly resolution (Weedon et al., 2014).
The dataset was interpolated linearly from its original 0.5° resolution to the MRB model resolution of
0.125°. The high resolution forecasts of the Global Environmental Multiscale atmospheric model – GEM
(Côté et al., 1998b, 1998a; Yeh et al., 2002), and the Canadian Precipitation Analysis – CaPA (Mahfouf et
al., 2007) datasets, often combined as (GEM-CaPA), provide the most accurate gridded climatic dataset
for Canada. Unfortunately, these datasets are not available prior to 2002 when most of the permafrost
observations used for model evaluation are available. Wong et al. (2017) performed an inter-comparison
of precipitation estimates from several products against observed station data over Canada and found
that CaPA and WFDEI products are in good agreement with station observations.

## 3.6 Spinning up and Stabilization

We used the first hydrological year of the climate forcing (Oct 1979-Sep 1980) to spin up the model
repeatedly for 2000 cycles while monitoring the temperature and moisture (liquid and ice content)
profiles at the end of each cycle for stabilization. We checked that the selected year was close to average
in terms of temperature and precipitation compared to the WFDEI record (1979-2016). The start of the
hydrological year was selected because it is easier to initialize the first cycle at the end of summer when
there is no snow cover or frozen soil moisture content. Stabilization is assessed visually using various plots



as well as by computing the difference between each cycle and the previous one making sure the absolute
difference does not exceed 0.1° for temperature (which is the accuracy of measurement thermostats) and
0.01 for moisture for all layers in the profile. The aim is to determine the minimum number of cycles that
can be used to inform the MRB model development, as it is computationally very expensive to spin up the
whole MRB model for 2000 cycles. We then assessed the impact of running the model for the period 1980-
2016 after 50, 100, 200, 500, 1000, and 2000 spin-up cycles (using the first hydrological year) on the ALD,
ZOD, and the temperature envelopes at the three sites for selected years depending on the available
observations. We focused on temperature changes as we found moisture profiles to stabilize quickly.

## 4. RESULTS

### 4.1 Establishing Initial Conditions

Figure 7 shows the temperature profiles at the end of spinning cycles for a selected GRU (NL Forest) for
the three selected sites using the two suggested soil layering schemes. NL Forest is representative of the
vegetation at the selected thermal sites for the three studied basins (except HPC bog site). As expected,
the profile changes quickly for the first few cycles then tends to stabilize so that there is no significant
change after 100 cycles and sometimes less. Figure 8 shows the temperature of each layer for the same
cases as in Figure 7 versus the cycle number to visualize the change patterns between cycles. There are
some small oscillations indicating some numerical issues but they do not cause major differences for the
simulations. For some cases/layers, the temperature keeps drifting (mostly cooling) for several hundred
cycles before stabilizing (if it occurs). We note a few important things:
• Changes to the temperature of the bottom layer (TBOT) from the initial value are too small to
have any significance; this triggered further testing using different initial values and the impact on
stabilization were similar as shown in the next sections. We also checked the model behaviour for
shallower soil columns and found that the bottom temperature did change with spinning up
within a range that decreased as the total soil depth increased.
• SC2 gives much more stable results than SC1 with faster stabilization and less drifting for all cases
indicating the importance of the vertical discretization scheme
• For layers where the temperature is drifting, the difference between the temperature after 2000
and 100 cycles is usually within 1.0 K.

**Possible Position of Figure 7**



The temperature gradient from South to North is clear comparing the different sites as well as the impact of the deeper permafrost in the North on the faster stabilization of temperature at HPC. Stabilization takes generally longer for middle layers at JMR than for BWC or HPC. For the three sites, there is a change in the slope of the profile at the depth corresponding to SDEP showing the importance of this parameter for permafrost simulations. This is due to the change in soil thermal and hydraulic properties above and below SDEP as well as the change of the heat transfer mechanism to become purely conductive below SDEP (there is no moisture). Above SDEP, there is some role for convective heat transfer depending on the moisture content and state (frozen/unfrozen) which in turn depend on soil properties and organic content.

**Possible Position of Figure 8**

Given the above findings, the remainder of the results focus on SC2 only. Additionally, we considered different values for the bottom temperature based on site location and extrapolation of observed temperature profiles as it cannot be established through spinning-up. Ground temperature measurements rarely go deeper than 20m and thus we do not know whether they are changing or not. There are established strong correlations between near surface ground temperature and air temperature at the annual scale (e.g. Smith and Burgess, 2000) but the near surface ground temperature is taken just a few centimeters below the surface. We spin up the model at the three sites for 2000 cycles for a few cases and then use the initial conditions after a selected number of cycles to run a simulation for the period of record (1979-2016) and assess the differences for ALD, ZOD, and temperature profiles for selected years within that period. The sensitivity of the results to SDEP, TBOT, and the organic content/configuration will then be assessed using 100 spin cycles only.

## 4.2 Impact of Spinning up

Figure 9, Figure 10 and Figure 11 show the simulated ALD, ZOD and temperature envelopes (selected years) at the three study sites respectively using initial conditions after 50, 100, 200, 500, 1000, and 2000 spin-up cycles using SC2 and the stated configuration for SDEP, TBOT, and ORG. Most differences are negligible and it is not easy to distinguish the different lines on those figures except for JMR where there are some larger differences in ALD and ZOD for some years depending on the initial conditions used. Assuming that more spinning up get us closer to the correct values, and thus considering the results initiated after 2000 cycles as a benchmark, one can accept an error of a few centimeters in simulated ALD with a smaller number of spin-up cycles. For JMR, this error is about 10% on average, which is much



smaller than the error in estimating ALD at this site. We are thus trading computational time for a slight
loss of accuracy at some sites, particularly those located in the more challenging sporadic zone.

**Possible Position of Figure 9**

The figures also include relevant observations to assess the quality of simulations. The simulated ALDs at
JMR and HPC are generally over-estimated (Figure 9). For HPC, two configurations are displayed: one with
mineral soil that has 18% organic matter for the top 0.6m (denoted M-org), which seems to better
represent the conditions at 01TC02; the other has a fully organic soil for the same depth (denoted ORG)
which results in a much smaller ALD and is closer to the thaw tube measurements at HPC (93-TT-02). This
indicates the large heterogeneity of conditions that can occur in close proximity of each other.
Temperature profiles are only shown for the first case as there are no observed temperature at the HPC
thaw tube site. For BWC, the ALD simulation is close to the observations for most years but the simulation
shows more inter-annual variability while observations show a small upward trend after an initial period
of large increase (1988-1992) which may be the result of the disturbance of establishing the site. A couple
of observations are marked "extrapolated" as the zero isotherm falls above the first thermistor (located
1m deep).

**Possible Position of Figure 10**

The simulated ZOD (Figure 10) is also over-estimated for JMR while it is close to values deduced from
observations for BWC and HPC. In contrast to ALD, observations have larger inter-annual variability than
simulation, possibly due to the large spacing of measuring thermistors and the failure of some in some
years. For HPC, the fully organic configuration (ORG) is showing more variability than the mineral one (M-
org) but both match the depth deduced from observations for 01TC02. In general, matching ZOD to
observations is not an objective in itself but its occurrence well within the selected soil depth is more
important. The largest value simulated is about 23m for HPC, which is less than half the total soil depth.
That indicates that a smaller soil column depth would not be recommended for HPC but could be used for
JMR and BWC.

**Possible Position of Figure 11**

Comparing to the observed envelopes at each site (Figure 11), the simulations look satisfactory in general.
The overall shapes of the profiles are captured for JMR and HPC despite the general over estimation of
ALD for both sites. At BWC, the active layer depth simulation agrees well with observations but the
temperature envelopes are generally colder than observed and gets the minimum envelope gets too cold





near the surface. A similar issue happens for JMR. This is not the case for HPC despite it being the coldest
site. This turned out to be related to the specification of fully organic soils at JMR and BWC while the
envelopes shown for HPC are taken from the mineral configuration that uses 18% organic content. This is
discussed further in Section 4.5.

## 4.3 Impact of Depth to Bedrock (SDEP)

SDEP for the above mentioned configurations for each site was perturbed in the range of 5-15m keeping
other studied parameters (TBOT and organic configuration) fixed. Figure 12 and Figure 13 show the impact
for each site on the average ALD and ZOD over the analysis period (1980-2016) for all land cover types.
100 spinning-up cycles were used to initialize those simulations and GRUs vary between the sites. For
JMR, wetlands do not develop permafrost while at shallower SDEP values, talik formations (i.e. no
permafrost) develop in some years and thus the shown averages on Figure 12 are for those years when
the soil is frozen all year round. There is a general tendency for ALD to decrease with deeper SDEP values
for all land cover types, especially for fully organic soils (JMR, BWC, and HPC ORG configuration). SDEP has
a similar impact on ZOD (Figure 13) for HPC, as the latter seems to decrease with deeper SDEP, but the
impact is not the same for BWC and JMR where ALD initially increases/decreases for JMR, BWC
respectively then becomes insensitive to SDEP. This possibly depends on the organic configuration. ZOD
is generally shallower for JMR followed by BWC and then HPC. Thus, this behaviour might be correlated
to the thickness of permafrost that increases in the same order.
**Possible Position of Figure 12**
**Possible Position of Figure 13**
Figure 14 shows how these changes to ALD and ZOD are occurring via changes in the shape of the
temperature envelopes. Increasing SDEP actually allows more cooling of the middle soil layers (between
0.5 – 10m) which pushes the maximum envelop upwards reducing ALD. The envelopes bend again to reach
the specified bottom temperature, which is much clearer for JMR (because it is set to +0.80°C) than BWC
and HPC where it is set to a negative value. Differences are larger for HPC for the fully organic soil
configuration (ORG) compared to the mineral configuration with 18% organic content (M-org). The
straighter envelopes of HPC tend to meet (i.e. at ZOD) at larger depths than the curved ones at BWC and
JMR. This cooling effect is possibly related to having moisture in deeper soil layers with deeper SDEP,
which affects the thermal properties of the soil as well as induces convective heat transfer.
Possible Position of Figure 14



## 4.4 Impact of Bottom Temperature (TBOT)

As shown by the spinning-up experiments above, the initial temperature of the deepest layer remains
virtually unchanged through the spin-up and thus has to be specified. The bottom of soil column has a
zero flux boundary condition (Section 3.3) implying no gradient at the bottom while TBOT is only an initial
condition that was expected to converge to a possibly different steady state value at the end of spin-up.
Temperature observations as deep as 50m are rare and relationships between that temperature and air
or near surface soil temperature are neither available nor appropriate. For the studied sites, it has been
estimated from the observed profiles, and perturbed within a range (-3.0 to +1.5°C), which was varied
depending on the site condition/location. Figure 15 shows the impact on changing the temperature of the
deepest layer on ALD while Figure 16 shows the impact on ZOD. For JMR, increasing TBOT increases ALD
quickly so that taliks form under wetlands if TBOT > 0°C and other land cover types follow at higher
temperatures such that permafrost does not develop under most canopy types if TBOT > 1.5°C. This gives
a way to simulate the no permafrost conditions observed at all sites in the basin (except 85-12B-T4). A
similar relationship is simulated for BWC as increasing TBOT increases ALD especially for wetlands. ALD at
HPC seems little affected by the bottom temperature with either organic configuration because of the
generally colder conditions. ZOD is showing low sensitivity to TBOT except for wetlands at JMR.
**Possible Position of Figure 15**
**Possible Position of Figure 16**
Figure 17 shows how the temperature envelopes respond to changes in TBOT. In all cases, the envelopes
seem to bend at some depth to try to reach the given bottom temperature. SDEP seems to influence the
start of that inflection. This bending towards the given temperature causes another inflection of the
maximum envelope closer to the surface. Depending on the depth of that first inflection, ALD may or may
not be affected. ZOD is not affected as much but the temperature at ZOD depends on TBOT. There is a
noticeable difference at HPC between the fully organic configuration (ORG) and the mineral configuration
that has 18% organic content (M-org) with the same depth (0.6m).
**Possible Position of Figure 17**

## 4.5 Impact of Organic Depth (ORG) and Configuration

It is believed that organic soils provide insulation to the impacts of the atmosphere on the soil
temperature, which would lead to a thinner active layer than the case of a fully mineral soil. This
assumption has been tested for the three sites by changing the depth of the fully organic layers (for JMR



and BWC) as well as against a mineral soil with relatively high organic content at HPC. The results are
sometimes counter-intuitive. Peat plateaux are widespread in the JMR region and thus the fully organic
layers are followed by layers of high organic content (50%) till SDEP. Increasing the fully organic layers
initially reduces ALD (Figure 18 top) as expected but also reduces ZOD (Figure 18 bottom) quickly. Then
the ALD (which is defined mainly by the maximum temperature envelop) increases again which means
that more fully organic layers provides less insulation than mineral layers with high organic content. The
reason may be related to the larger moisture holding capacity provided by fully organic layers or because
the sand content is small and thus the hydraulic conductivity of the mineral layers is low. HPC shows a
similar behaviour where 3 organic layers have a similar effect on ALD as 6 layers and the minimum ALD is
reached by 4-5 layers. BWC has a different behaviour than the other two sites as ALD increases initially
when increasing the fully organic layers from 3 to 4 then decreases gradually. ZOD seems to decrease with
increasing the organic depth for most land cover types at the three sites. Wetlands behave in a different
way compared to other land cover types at the different sites because it is configured to remain close to
saturation as much as possible. At JMR, wetlands are not underlain by permafrost for all organic
configurations, which agrees with the literature.

**Possible Position of Figure 18**

Figure 19 shows the response of the temperature envelopes to changes in the organic depth. Increasing
the organic depth causes much larger negative temperatures near the surface for the minimum envelope
but causes the inflection of the minimum envelop to occur at slightly higher temperatures. A similar effect
can be seen for the maximum envelop. The maximum envelopes for the different organic depth intersect,
which corroborates with the above for ALD. Another interesting feature can be observed comparing the
ORG and M-org configurations for HPC in Figure 14 and Figure 17. The M-org configuration has a much
smaller temperature range near the surface than the fully organic soil and causes less cooling in the
intermediate soil layers (above SDEP) such that the observed profiles are better matched for this site.
These results emphasize the need to investigate the soil hydraulic and thermal properties for each case
to better understand the role of organic matter and fully organic layers on the moisture and temperature
simulations.

**Possible Position of Figure 19**



## 5. Discussion and Conclusions

Permafrost is an important feature of cold regions, such as the Mackenzie River Basin, and needs to be
properly represented in land surface hydrological models, especially under the unprecedented climate
warming trends that have been observed. The current generation of LSMs are being improved to simulate
permafrost dynamics by allowing deeper soil profiles than typically used and incorporating organic soils
explicitly. Deeper soil profiles have larger hydraulic and thermal memories that require more effort to
initialize. We followed the recommendations of previous studies to select the total soil column depth to
be around 50m. The temperature envelopes meet well within the 50m soil column over the simulation
period (including spinning-up), i.e. the bottom boundary condition is not disturbing the simulated
temperature profiles/envelopes and ALD.
We analysed the conventional layering schemes used by other LSMs, which tend to use an exponential
formulation to maximize the number of layers near the surface and minimize the total number of layers.
We found that the exponential formulation is not adequate to capture the dynamics of the active layer
depth and thus tested two other alternative schemes that have smaller thicknesses for the first 2 meters,
instead of the conventional exponentially increasing thicknesses. The first scheme (SC1) had equally-sized
layers in the first 1m, followed by thicker but equally-sized layers in the second 1m. The second scheme
(SC2) was formulated to have increasing thicknesses with depth following a scaled power law, which we
found to be more suitable for the explicit forward numerical solution used by CLASS.
We discussed the common initialization approaches, including spinning up the model repeatedly using a
single year or a sequence of years, spinning up the model in a transient condition on long paleo-climatic
records, or combining both of these approaches. Paleo-climatic reconstructions are scarce and provide
limited information (e.g. mean summer temperature or total annual precipitation), while LSMs typically
require a suite of meteorological variables at a high temporal resolution for the whole study domain.
These variables can be stochastically generated at the resolution of interest informed by paleo-records.
However, such practice is computationally expensive, especially for large domains and also introduces
additional uncertainties. The approach of spinning-up using available 20[th] century data has been criticized
as picking up the anthropogenic climate warming signal that started around 1850 and thus would yield
initial conditions that are not representative. However, paleo climatic records also show that the climate
has always been transient and there may not exist a long enough period of quasi-equilibrium to start the
spinning-up process (Razavi et al., 2015). Spinning-up using a sequence of years is thus more prone to
having a trend than a single year and de-trending the sequence is not free of assumptions either.





Given the above complications, we investigated the impact of the simplest approach, which is spinning-
up using a single year, on several permafrost metrics (active layer depth – ALD, zero oscillation depth
where the temperature envelopes meet – ZOD, and annual temperature envelopes). The aim was to
determine the minimum number of spinning-up cycles to have satisfactory performance (if reached) and
to know how much accuracy is lost by not spinning more. We did this for three sites along a south-north
transect in the Mackenzie River Valley sampling the different permafrost zones (sporadic, extensive
discontinuous and continuous) in order to be able to generalize the findings to the whole MRB domain.
Additionally, we investigated the sensitivity of the results to some important parameters such as the
depth to bedrock (SDEP), the temperature of the deepest layer (TBOT), and the organic soil configuration
(ORG).
The results show that temperature profiles at the end of spinning cycles remained virtually unchanged
(i.e. reached a quasi steady state) after 50-100 cycles, when benchmarked against the results of 2000
cycles. We focused on temperature for this stability analysis, because we found that the soil moisture
profiles (both liquid and frozen) stabilize much earlier during spin-up. In some cases, changes in the middle
layers occurred after 100 cycles but the influence of that on the simulated envelopes, ALD and ZOD was
found to be small to negligible compared to the uncertainty of observations and the scale of our model.
We also found that the selection of the layering scheme has an effect on stabilization and our proposed
scheme (SC2) with increasing thicknesses with depth reached stability faster and had less drifting.
Therefore, the simple single-year spinning approach seems to be sufficient for our purpose using SC2.
We also found that the temperature of the deepest soil layer (TBOT) remained virtually unchanged from
the specified initial value even after 2000 spinning cycles. Therefore, this temperature has to be specified
by the modeller. For the study sites, we extrapolated it from the observed envelopes and studied the
effect of perturbing it around the extrapolated value. This perturbation had small impacts on ALD and
ZOD except for JMR in the sporadic zone, but it had a significant impact on the shape of the envelopes.
Temperature observations going as deep as 50m are rare. Most of the permafrost monitoring sites in the
MRB have up to 20m cables and thus we do not know if temperature of deeper soil layers has been
changing over time, and if so, by how much. To take the information back to MRB scale, we recommend
using a south to north gradient moving from +1.0 in the sporadic zone to -2.0 in the continuous zone and
specifying a spatially variable field as an input initial condition. For this study, we considered only the zero-
flux boundary condition. It is possible to test whether a non-zero thermal flux boundary condition could




resolve this issue. However, available datasets for the geothermal flux are not transient and estimate
those fluxes at depths greater than the 50m used and thus the issue may need further investigation.
The analyses also demonstrated the importance of the organic soil configuration (i.e. how many layers
and their organic sub-types) and depth to bedrock on the simulated temperature profiles and active layer
dynamics. In most cases, we found combinations of TBOT, SDEP, and ORG that produced satisfactory
simulations but the impact of organic layering seems to require further investigation, as increasing the
thickness of organic layers does not always act to reduce ALD or reduce the cooling in the middle soil
layers that should result from increased insulation. There is an interplay between the moisture
properties/content and thermal properties of organic soils that needs further investigation. Additionally,
we cannot represent mixed canopies using CLASS, e.g. trees or shrubs underlain by moss. Moss could be
providing additional insulation under those canopies that is not represented.
To conclude, we now have an approach to represent permafrost in MESS/CLASS at the MRB that has the
following features:
- Around a 50m deep soil profile with increasing soil thickness with depth
- Spinning 50-100 cycles of the first year of record to initialize the moisture and temperature
profiles
- TBOT, SDEP, and soil texture parameters are to be specified spatially. We have processed gridded
data for SDEP and soil texture (including organic matter) and modified MESH/CLASS to read these
by grid. In preparing these fields, we will use the 30% threshold to activate fully organic soils.
It was necessary to increase the flexibility of the MESH framework to accommodate these input formats
as well as to produce relevant permafrost outputs. However, the model is still deficient in some ways. For
example, the explicit forward numerical solution may be limiting our choices for soil layering and the lack
of complex canopies, amongst other things, may be affecting our parameterization of MESH. These
findings are not specific to MESS/CLASS and could be beneficial for the LSM community. This study also
demonstrated a simple and effective way to use small-scale investigations to inform larger scale
modelling. The key is to use the same model at both scales.





## Acknowledgements

This research was undertaken as part of the Changing Cold Region Network, funded by Canada's Natural
Science and Engineering Research Council and by the Canada Excellence Research Chair in Water Security
at the University of Saskatchewan.

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





# Figures

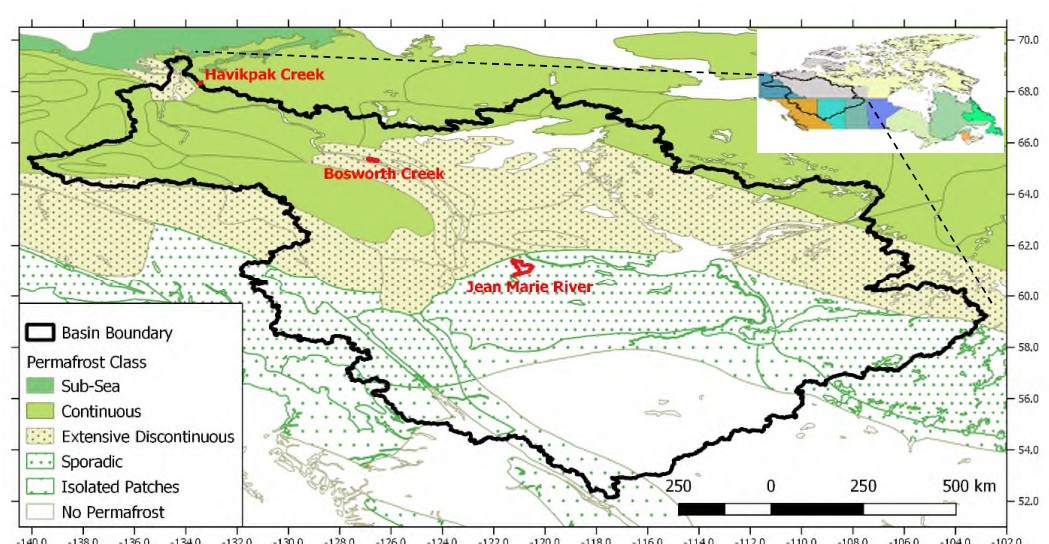


*Figure 1 Mackenzie River Basin: Location, Permafrost Classification, and the Three Study Sites*

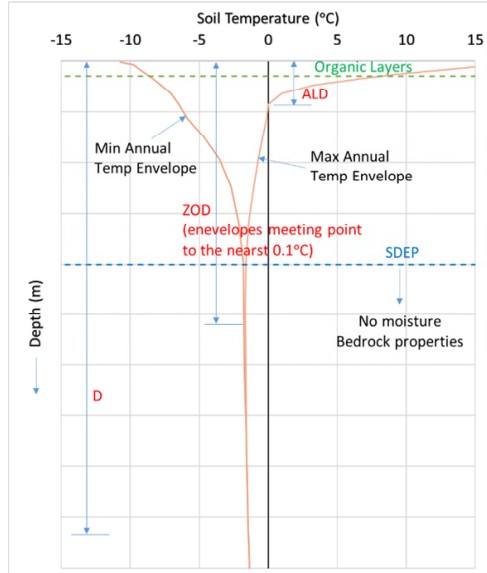


*Figure 2 Schematic of the Soil Column showing the Main Variables used to Study Permafrost*





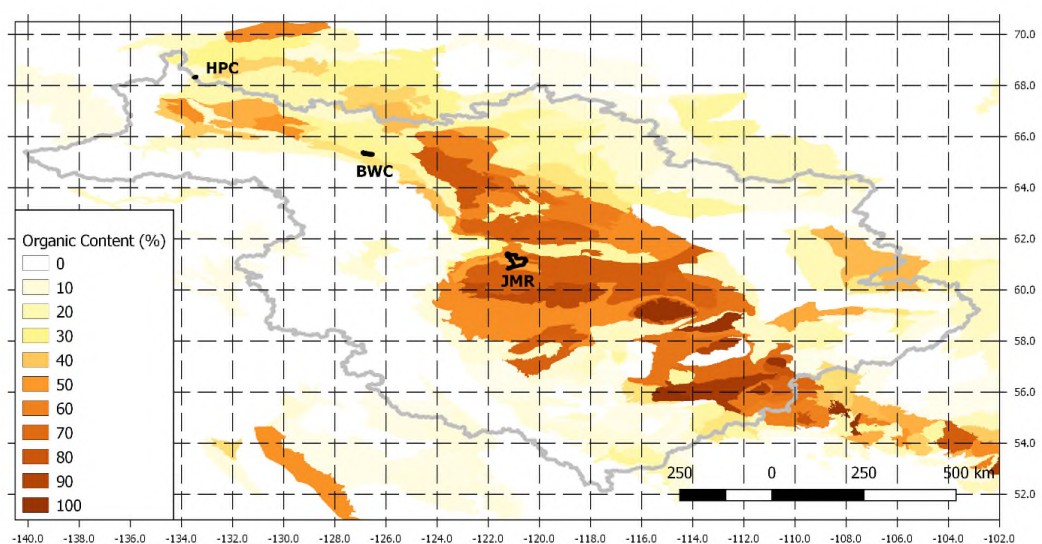


*Figure 3 Processed Percentage of Organic Matter in Soil at 0.125° from SLC v2.2 Dataset (Centre for Land and Biological Resources Research, 1996)*

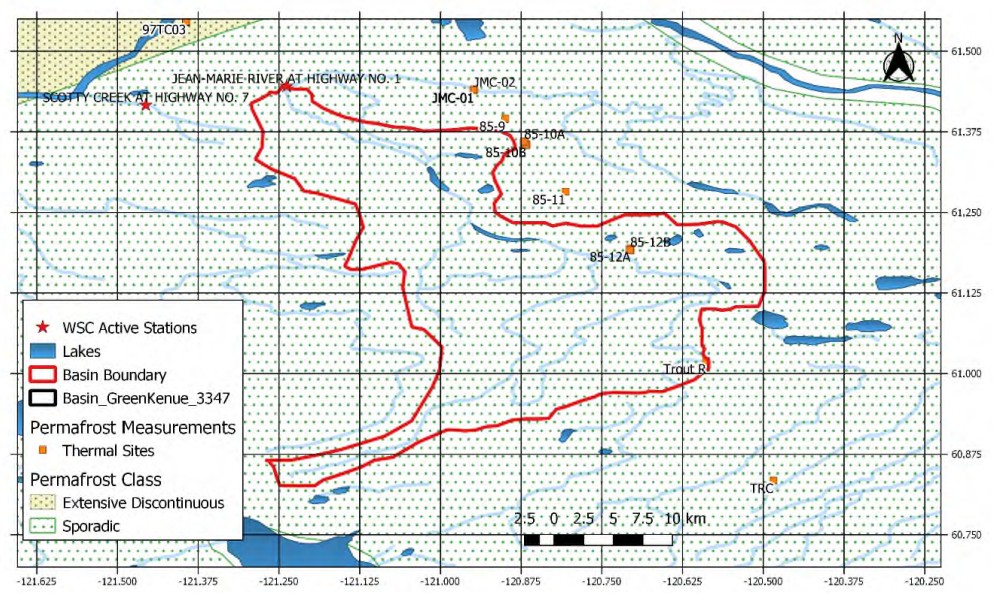


*Figure 4 Permafrost Measurement Sites around Jean Marie River*



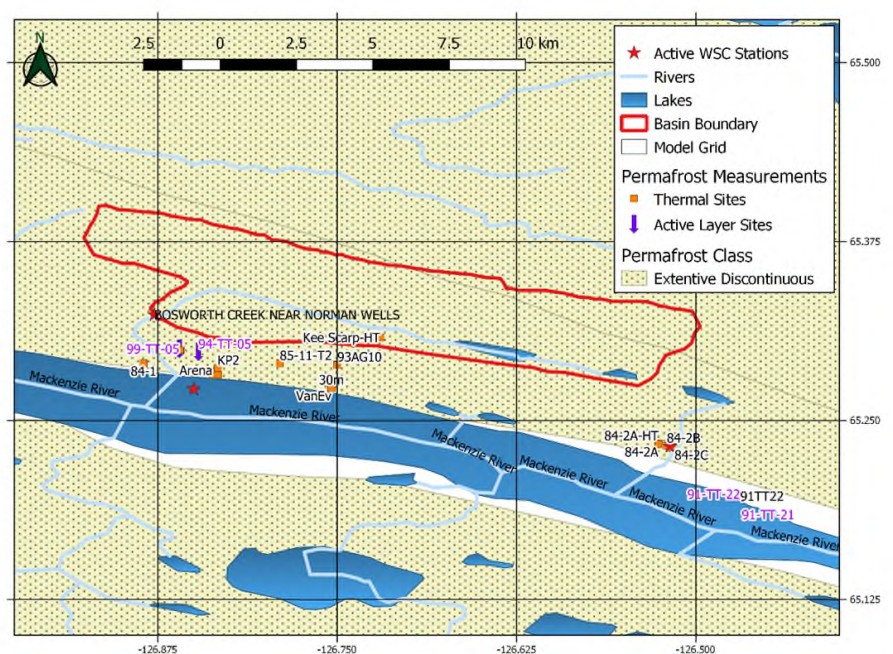


*Figure 5 Permafrost Measurement Sites around Bosworth Creek*

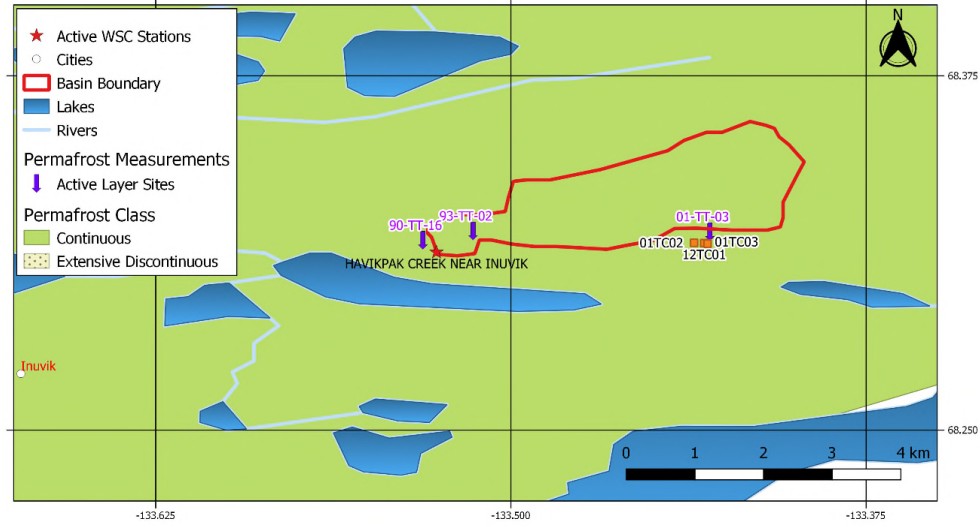


*Figure 6 Permafrost Measurement Sites around Havikpak Creek*








*Figure 7 Temperature Profiles at the End of a Range of Spin-up Cycles for NL Forest
at the Three Study Sites using Different Soil Layering Schemes*

*Figure 8 Impact of Soil Layering Scheme Selection on Spin-up Convergence at the Three Study Sites (the
darker the color, the deeper the layer, deepest layer is colored blue)*




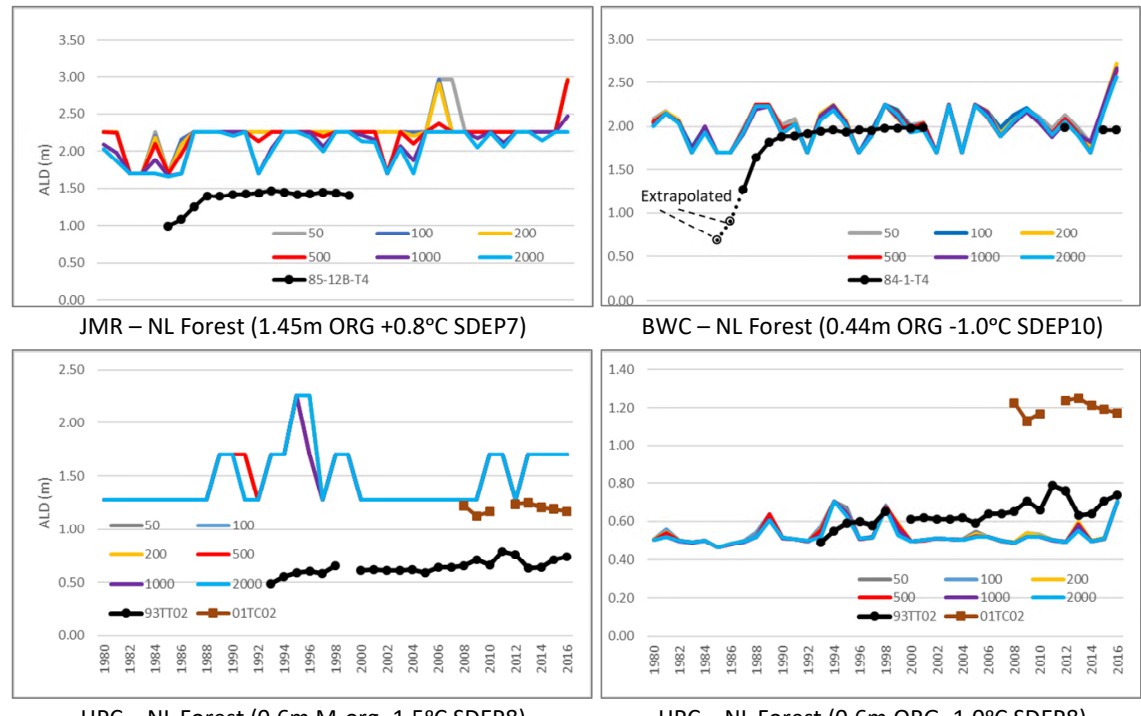

*Figure 9 Impact of Number of Spin-up Cycles on Simulated ALD for Needle Leaf Forest Tiles at the Three Study Sites – 2 organic configurations used for HPC*




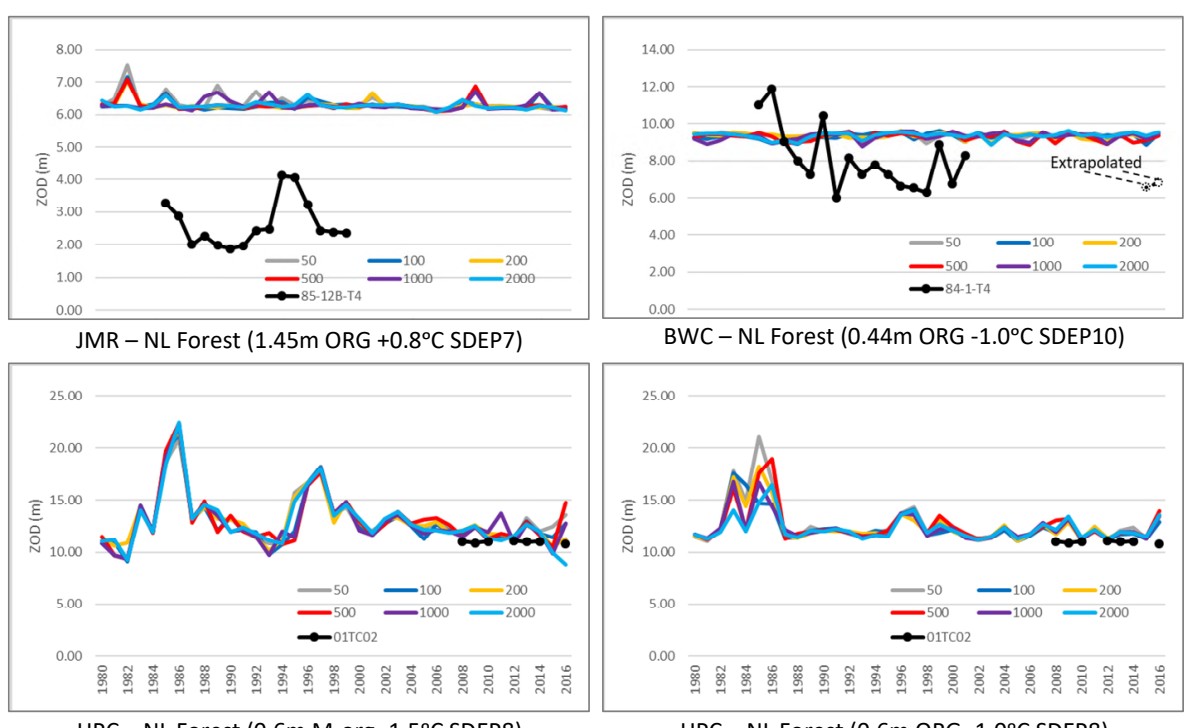

*Figure 10 Impact of Number of Spin-up Cycles on Simulated ZOD for Needle Leaf Forest Tiles at the Three Study Sites – 2 organic configurations used for HPC*



*Figure 11 Impact of Number of Spin-up Cycles on Simulated Temperature Envelopes for Needle Leaf Forest Tiles for a Selected Year at Each Study Site (M-org configuration is shown for HPC)*





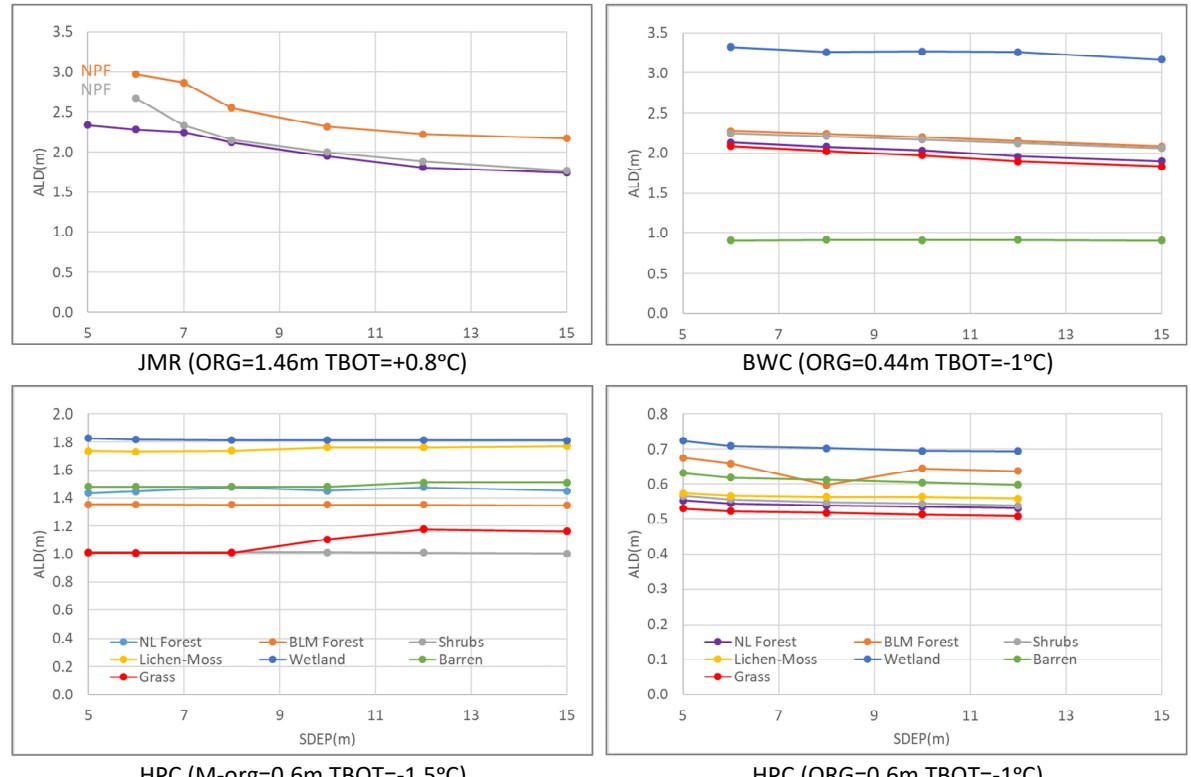

*Figure 12 Impact of SDEP on Average Simulated ALD for Different GRUs at the Three Study Sites over the 1980-2016 Period – 2 organic configurations used for HPC*



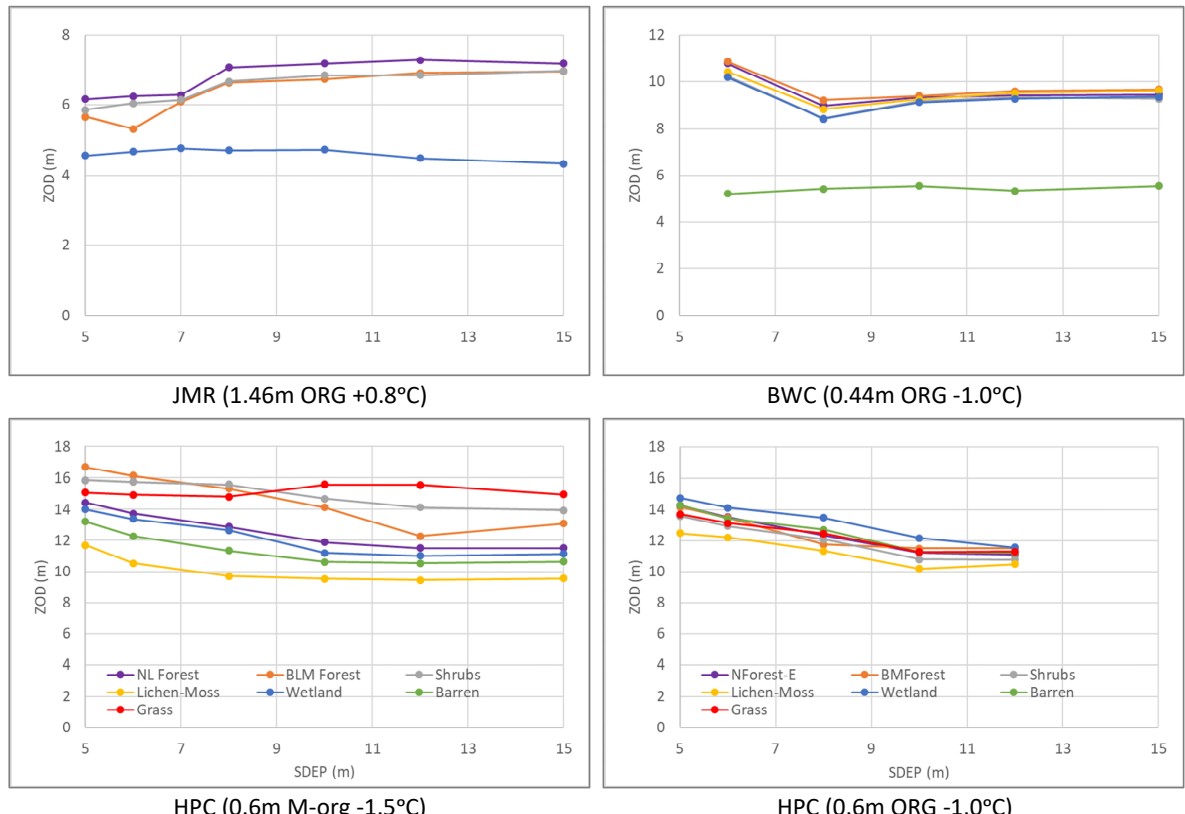

Figure 13 Impact of SDEP on Average Simulated ZOD for Different GRUs at the Three Study Sites over the 1980-2016 Period – 2 organic configurations used for HPC





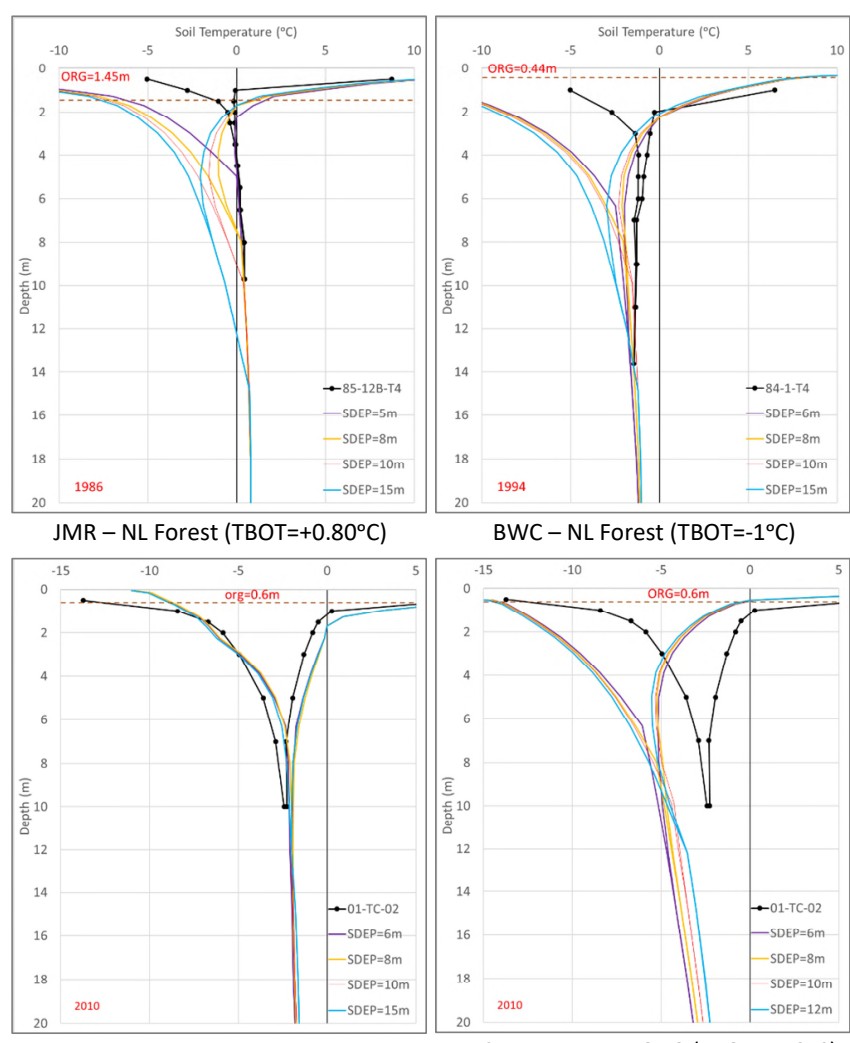

JMR – NL Forest (TBOT=+0.80°C)     BWC – NL Forest (TBOT=-1°C)

HPC – NL Forest – M-org (TBOT=-1.5°C)     HPC – NL Forest – ORG (TBOT=-1.0°C)

*Figure 14 Impact of SDEP on Simulated Temperature Envelopes for Needle Leaf
Forest Tiles for a Selected Year at Each Study Site – 2 organic configurations are
used for HPC*





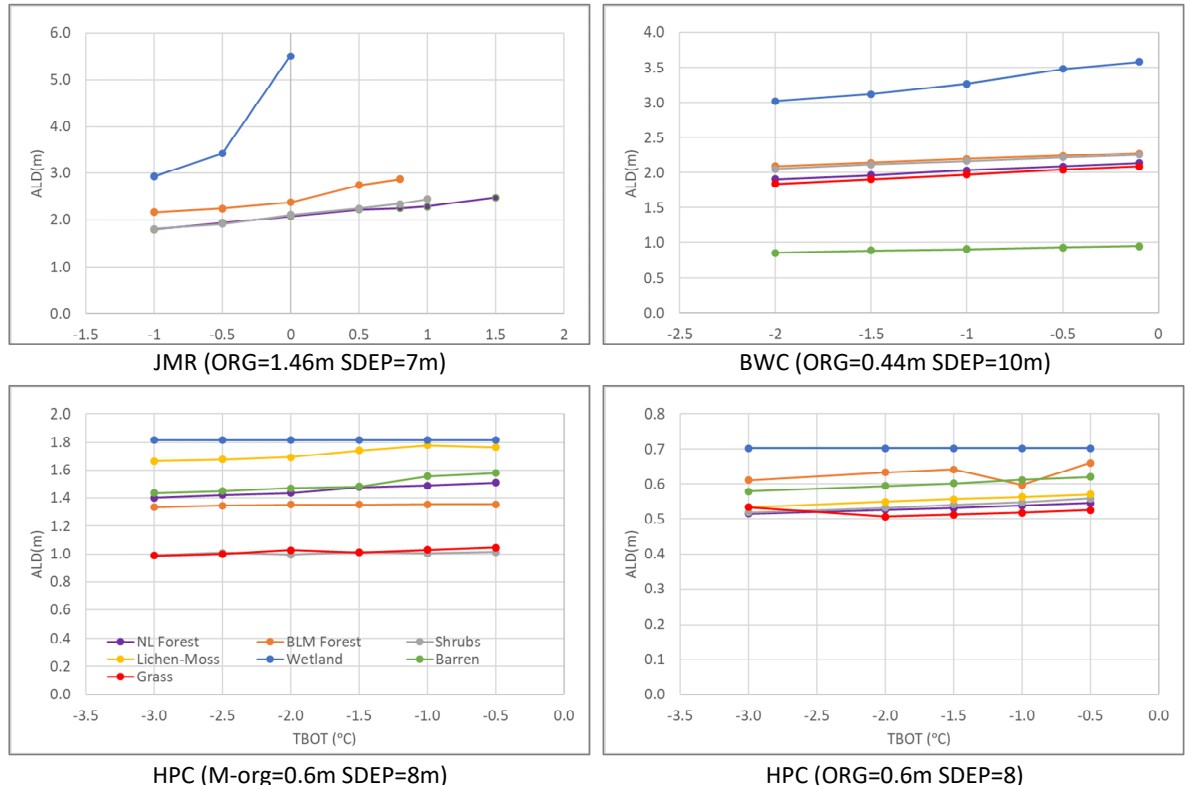

*Figure 15 Impact of TBOT on Average Simulated ALD for Different GRUs at the thee sites over the 1980-2016 period – 2 organic configurations used for HPC*





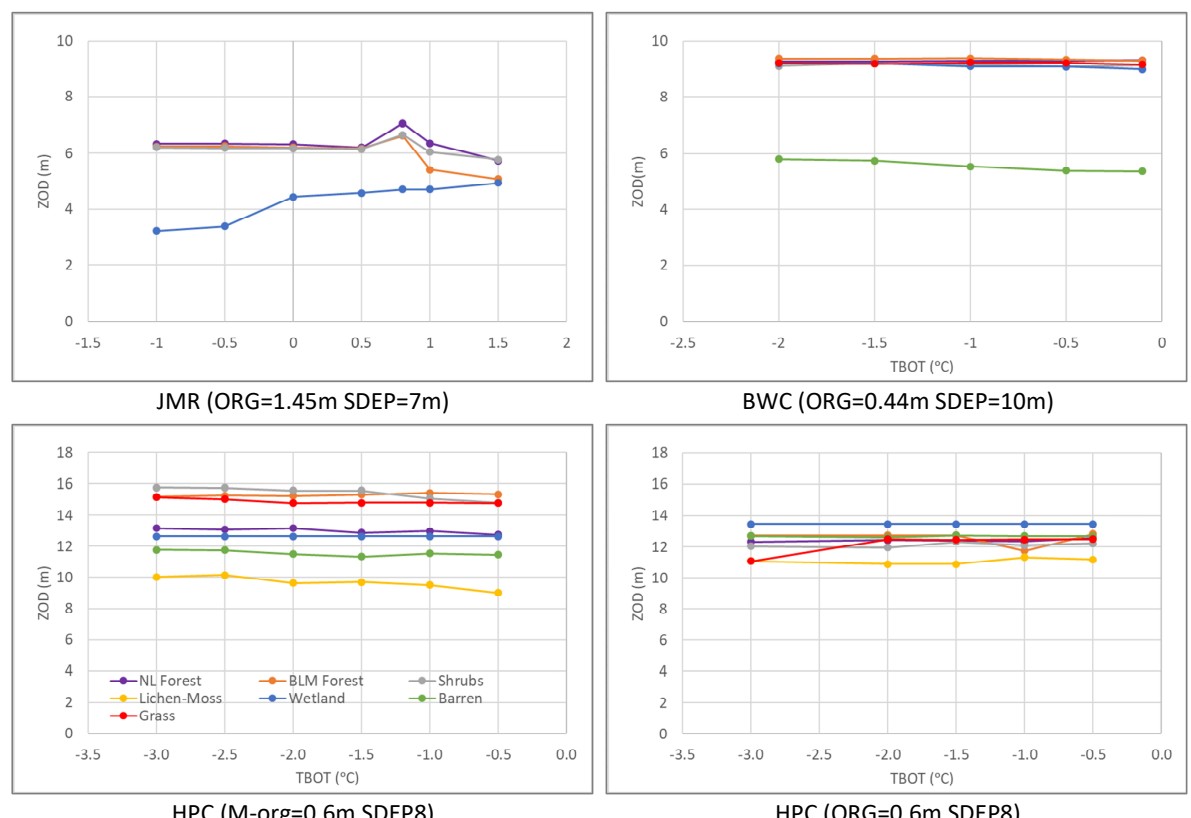

*Figure 16 Impact of TBOT on Average Simulated ZOD for Different GRUs at the Thee Study Sites over the 1980-2016 Period – 2 organic configurations used for HPC*





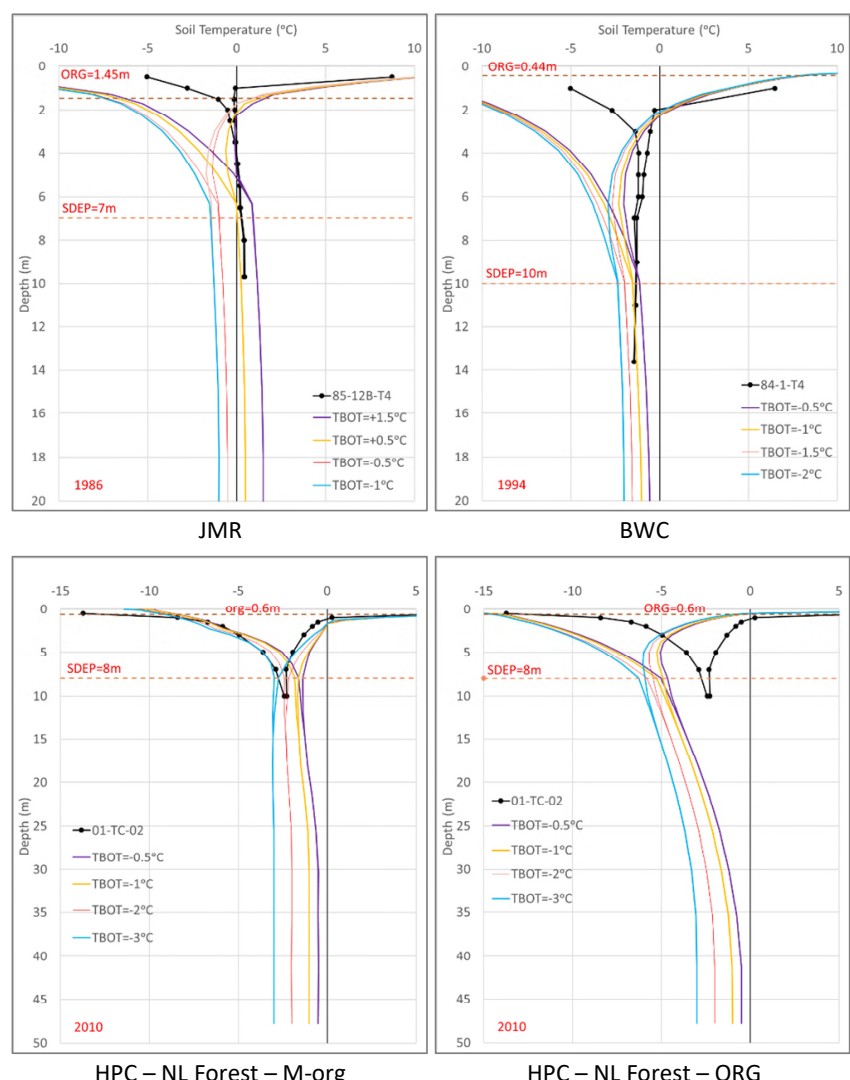

*Figure 17 Impact of TBOT on Simulated Temperature Envelopes for Needle Leaf Forest Tiles for a Selected Year at each Study Site – 2 organic configurations are used for HPC*



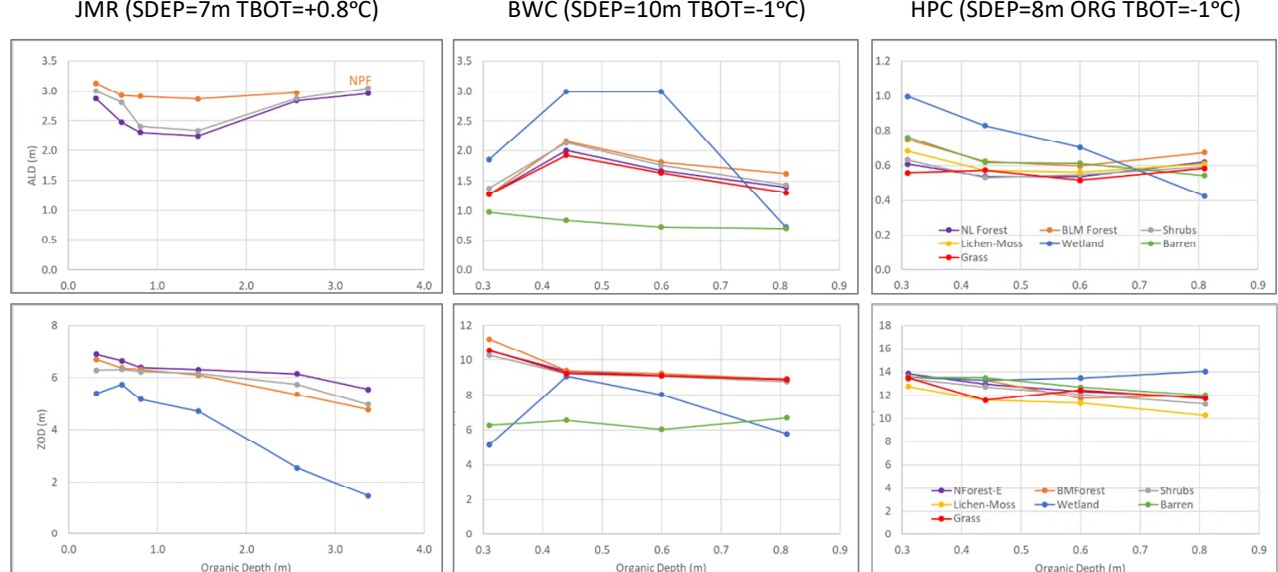

*Figure 18 Impact of Organic Depth on Average (1980-2016) Simulated ALD and ZOD for Different GRUs
at the Three Study Sites*



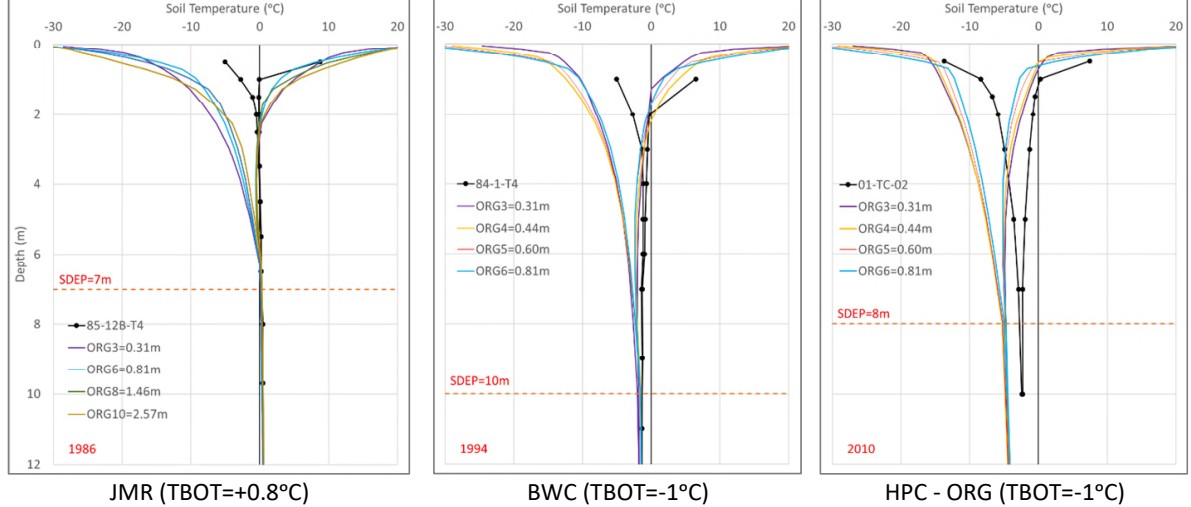

*Figure 19 Impact of  Organic Soil Depth on Simulated Temperature Envelopes for Needle Leaf Forest Tiles for a Selected Year at Each Study Site*





# Tables

*Table 1 Permafrost Sites and Important Measurements for Study Basins*

| Site Name | Site ID | Type | Cables (Depth in m) | Data* | Vegetation | Permafrost Condition |
|---|---|---|---|---|---|---|
| **JMR (Fort Simpson)** | | | | | | |
| Jean-Marie Creek | JMC-01 | Thermal | T1 (5) | 2008-2016 | Shrub Fen | No |
| | JMC-02 | Thermal | T1 (5) | 2008-2016 | Needle Leaf Forest | No |
| Pump Station 3 | 85-9 (NWZ9) | Thermal | T1 (5), T2 (5), T3 (20), T4 (20) | 1986-1995, 2012-2016 | Needle Leaf Forest/Shrubs/ Moss | No |
| Jean Marie Creek A | 85-12A | Thermal | T1 (5), T2 (5), T3 (16.4), T4 (12) | 1986-1995 | | No |
| Jean Marie Creek B | 85-12B (NWZ12) | Thermal | T1 (5), T2 (5), T3 (17.2), T4 (9.7) | 1986-2000 | | Yes |
| Mackenzie Hwy S | 85-10A | Thermal | T1 (5), T2 (5), T3 (20), T4 (20) | 1986-1995 | N/A | No |
| | 85-10B | Thermal | T1 (5), T2 (5), T3 (10.5), T4 (10.5) | 1986-1995 | N/A | No |
| Moraine South | 85-11 | Thermal | T1 (5), T2 (5), T3 (12), T4 (12) | 1986-1995, 2014-2016 | N/A | No |
| **BWC (Norman Wells)** | | | | | | |
| NW Fen | 99-TT-05 | Thaw Tube | | 2009 | Needle Leaf Forest/Moss | Yes |
| | 99-TC-05 | Thermal | Near Surface | 2004-2008 | | |
| Normal Wells Town | Arena | Thermal | T1 (16) | 2014-2015 | Disturbed area adjacent to parking lot | Yes |
| | WTP | Thermal | T1 (30) | 2014-2017 | | Yes |
| KP 2 - Off R.O.W. | 94-TT-05 | Thaw Tube | | 1995-2007 | Needle Leaf Forest/Shrubs/ Moss | Yes |
| Norman Wells (Pump Stn 1) | 84-1 | Thermal | T1 (5.1), T2 (5), T3 (10.4), T4 (13.6), T5 (19.6) | 1985-2000 1985-2016 | | Yes |
| Van Everdingen | 30m | Thermal | T1 (30) | 2014-2017 | Needle Leaf /Mixed Forest | Yes |
| Kee Scrap | Kee Scrap-HT | Thermal | T1 (128) | 2015-2017 | Mixed Forest | No |
| **HPC (Inuvik)** | | | | | | |
| Havikpak Creek | 01-TT-02 | Thaw Tube | | 1993-2017 | Needle Leaf Forest | Yes |
| Inuvik Airport | 01-TT-03 | Thaw Tube | | 2008-2017 | | Yes |
| Inuvik Airport | 90-TT-16 | Thaw Tube | | 2008 | | Yes |
| Upper Air | 01-TT-02 | Thaw Tube | | 2008-2017 | N/A | Yes |
| Inuvik Airport (Trees) | 01-TC-02 | Thermal | T1 (10) | 2008-2017 | Needle Leaf Forest | Yes |
| Inuvik Airport (Bog) | 01-TC-03 | Thermal | T1 (8.35) | | Wetland | Yes |
| | 12-TC-01 | Thermal | T1 (6.5) | 2013-2017 | | Yes |





*Table 2 Soil Layering Schemes*

| | First Scheme (SC1) | | | Second Scheme (SC2) | | |
|---|---|---|---|---|---|---|
| Layer | Thickness | Bottom | Center | Thickness | Bottom | Center |
| 1 | 0.10 | 0.10 | 0.05 | 0.10 | 0.10 | 0.05 |
| 2 | 0.10 | 0.20 | 0.15 | 0.10 | 0.20 | 0.15 |
| 3 | 0.10 | 0.30 | 0.25 | 0.11 | 0.31 | 0.26 |
| 4 | 0.10 | 0.40 | 0.35 | 0.13 | 0.44 | 0.38 |
| 5 | 0.10 | 0.50 | 0.45 | 0.16 | 0.60 | 0.52 |
| 6 | 0.10 | 0.60 | 0.55 | 0.21 | 0.81 | 0.71 |
| 7 | 0.10 | 0.70 | 0.65 | 0.28 | 1.09 | 0.95 |
| 8 | 0.10 | 0.80 | 0.75 | 0.37 | 1.46 | 1.28 |
| 9 | 0.10 | 0.90 | 0.85 | 0.48 | 1.94 | 1.70 |
| 10 | 0.10 | 1.00 | 0.95 | 0.63 | 2.57 | 2.26 |
| 11 | 0.20 | 1.20 | 1.10 | 0.80 | 3.37 | 2.97 |
| 12 | 0.20 | 1.40 | 1.30 | 0.99 | 4.36 | 3.87 |
| 13 | 0.20 | 1.60 | 1.50 | 1.22 | 5.58 | 4.97 |
| 14 | 0.20 | 1.80 | 1.70 | 1.48 | 7.06 | 6.32 |
| 15 | 0.20 | 2.00 | 1.90 | 1.78 | 8.84 | 7.95 |
| 16 | 1.00 | 3.00 | 2.50 | 2.11 | 10.95 | 9.90 |
| 17 | 2.00 | 5.00 | 4.00 | 2.48 | 13.43 | 12.19 |
| 18 | 3.00 | 8.00 | 6.50 | 2.88 | 16.31 | 14.87 |
| 19 | 4.00 | 12.00 | 10.00 | 3.33 | 19.64 | 17.98 |
| 20 | 6.00 | 18.00 | 15.00 | 3.81 | 23.45 | 21.55 |
| 21 | 8.00 | 26.00 | 22.00 | 4.34 | 27.79 | 25.62 |
| 22 | 10.00 | 36.00 | 31.00 | 4.90 | 32.69 | 30.24 |
| 23 | 14.00 | 50.00 | 43.00 | 5.51 | 38.20 | 35.45 |
| 24 | | | | 6.17 | 44.37 | 41.29 |
| 25 | | | | 6.87 | 51.24 | 47.81 |

*Table 3 Number of Layers of Each Organic Sub-type for the Organic Configurations Used*

| | | Organic Sub-Type | | |
|---|---|---|---|---|
| Organic Configuration | Depth (m) SC2 | 1 (Fibric) | 2 (Hemic) | 3 (Sapric) |
| 3ORG | 0.31 | 1 | 1 | 1 |
| 4ORG | 0.44 | 1 | 1 | 2 |
| 5ORG | 0.60 | 1 | 2 | 2 |
| 6ORG | 0.81 | 2 | 2 | 2 |
| 8ORG* | 1.46 | 2 | 3 | 3 |
| 10ORG* | 2.57 | 3 | 3 | 4 |
| 11ORG* | 3.37 | 3 | 4 | 4 |

*Only used for JMR