# Peer review of "On the Configuration and Initialization of a Large Scale Hydrological Land Surface Model to Represent Permafrost"

_Hydrology and Earth System Sciences, 2019_

## Referee Comment (RC1) · Anonymous Referee #1 · 30 May 2019

General comments

Elshamy et al. detail testing and resultant guidelines for the configuration and initialization (especially 'spinning up') of permafrost in large-scale hydrologic models, with a focus in this study of the Mackenzie River Basin. Permafrost exerts primary control on hydrologic routing in cold regions, and thus this topic is critical for Canada and other countries with high latitude regions that are experiencing high rates of warming. As such, the manuscript scope is a good fit for HESS, and the collective authorship team offers many decades of modeling experience and insight. I think the paper will be a useful contribution to HESS, but I think it needs some reworking.

[Figure]

Major concerns

1. This is a vague concern, but this comes across as a bit of a high-end technical report in places, more than a research paper. Rather than detail why that is, I list a few specific concerns below that map to this general overarching theme.

2. This discussion on changing annual discharge is a bit overly simplified. I'd break this down a bit more into different seasons. There is a pretty consistent increase in minimum flows across the pan-Arctic (see, for example, the recent ECCC report, Canada's Changing Climate, or Walvoord and Striegl 2007 GRL, St Jacques and Sauchyn 2009 GRL, Duan et al. 2017 Water – for China)

3. The intro is quite long – it is 6 paragraphs, of which several are long. Also, the objectives section which follows is normally embedded in the intro in most papers. This would add about another 2 paragraphs. This needs to be trimmed. Paragraph 4 is especially wordy. Paragraphs 5 and 6 could be cut by 50%.

4. Lists or bulleted sections are not written very parallel in this paper, and they are hard to relate and read. L155-168, L459-467, and L686-691 are examples.

5. L205-210, this is very late in the paper to be delineating the focus

6. Because there are three sites, the site description is very long (Sections 3.2.1, 3.2.2, and 3.2.3. This takes up about 7 pages, which is similar in length to short paper on its own. Basically, some of this information (especially the inordinate focus on parameterization, when that is not the point of the study) needs to be moved to an electronic supplement. It detracts from the key messaging, and it's not a very invigorating read. I think the site description is key, but could be shorter, but I don't think the reader needs to wade through endless parameter justification, which could be built into tables in a supplement for interested readers.

7. Oct. 1979 is certainly late in history as a representative climate from which to base the model spin up. I realize this is briefly addressed later, but I suspect a permafrost

modeler would object.

8. I think a small section on the thermal physics in the model (governing equations, soil freezing curves if any, etc.) would be far more useful to the reader than the emphasis on parameters.

9. This contribution is very qualitative and even anecdotal in places. For example 'seems' should up 7 times in the manuscript, while 'seem' shows up in 8 places. The difference in model runs are not compared via standard metrics like RMSE or something like that. The discussion seems to rather focus on apparent discrepancies and vague explanations. For examples of this, just consider any section on model comparisons or differences. Also, note recurring appearances of 'much more' and 'too small' – a few actual numbers would be nice.

10. The authors do not frame their permafrost modeling results in the discussion around past contributions. Cryosphere scientists have been modeling permafrost and considering spin up scenarios for a very long time. The authors' work is new and interesting (especially the focus on the inclusion of permafrost in large-scale hydrologic modeling), but the thermal physics under consideration are not overly new, and it would make sense to relate their study findings

Minor concerns

Many of these are quite trivial

L13, comma after 'average'

L33, shouldn't basin be capitalized here as elsewhere when preceded by Mackenzie or Mackenzie River?

L33, 'heating up by 4 degC' . . ..over what time period? 100,000 years? 50 years?

L36, 'American rivers' should just be 'America' and the subsequent semicolon should be a comma

L75, 'implied' should be 'inferred'

L119, 'In addition to. . ..for spinning' is a fragment

L143 and elsewhere, 'etc.' occurs in the paper where it is entirely superfluous in a couple places. It tends to look choppy – 'such as' is sufficient.

L141, Land does not need to be capitalized

L152, does this mean that sandstone thermal properties are always used the for the bedrock conductivity everywhere? This seems less than ideal.

L161 'thus we use a thaw, rather than a freeze criterion' – I have no idea what this means

L167, This is more commonly called the 'seasonal penetration depth', at least in non-permafrost regions

L170, permafrost is not defined cryotically like this (frozen vs. unfrozen). It's a temperature definition – i.e. ground below 0C for two or more consecutive years – see, for example, Dobinski, 2011, Earth Science Reviews.

L173, "MESH/CLASS used to output" should change to 'Prior versions of MESH/CLASS outputted merely temperature profiles' or something like this

175, 'A CLASS typical' should be 'A typical CLASS'

L192, "these has" should be 'these have', and I'm not sure what 'to be carried back to the MRB scale' means

L197, 'North West' should be 'Northwest'

L216, 'with' should be 'by' and the rest of this sentence needs to be rewritten as it is confusing what it means

L220-221, Weather and North should not be capitalized

L302, should not be semicolon

L304, why does deep permafrost imply no groundwater? It would make more sense to note that the cold climate prevents the formation of a lateral talik, and thus there is no perennial shallow groundwater. See Lamontagne-Halle et al. 2018 (Environmental Research Letters) or Connon et al. 2018, JGR-Earth Surface.

L316 (and elsewhere, do a word search), 'envelops' should be 'envelopes'
* * *

---

## Referee Comment (RC2) · Anonymous Referee #2 · 29 Jul 2019

**Reviewer :**

This study aims to derive a robust, yet computationally efficient initialization parameterization approach that can be applied to regions where data are scarce and simulations typically require large computational resources. An upscaling approach to inform large-scale ESM simulations based on the insights gained by modelling at small scales was performed. The results show that the model has good performance in reproducing present-climate permafrost properties at the three sites at the Mackenzie River Valley. The results also demonstrate that the simulations are sensitive to the soil layering scheme, the depth to bedrock, and the organic soil properties.

It is really important to investigate the performances of hydrological and land surface models in permafrost regions under climate change. However, there are some shortcomings that might affect the contribution of this study. My main concern and comments are listed as follow.

**General comments**

1. Lin 34:...however, are not so clear…You should give citations.
2. Line 39: What do you mean "uncertainty"?
3. Line 46-50: You give importance of permafrost here, which may be not suite for this paragraph. I suggest that you provide separate paragraph to show the importance of permafrost and the progress in interaction between permafrost and hydrological at the beginning of the introduction.
4. Line 51 and 91: Here the authors give the modeling work in hydrological processes in permafrost regions, I noticed that the models were all land surface models. As I know, there were many modeling work that has been done by hydrological models in cold regions, such as VIC, GBHM. I would suggest that the authors to provide the different with hydrological models and land surface models on the previous modeling work in hydrological processes in cold regions, then clearly state why you choose the land surface model for this study.
5. Section 3.2 Study Sites and Data: This section is too long. Please make it concise using figures and tables. In addition, you may combine Section 3.5 (Climate Forcing) with this Section. They are all data introduction.
6. Line 170-171, Permafrost, which is defined as ground in which temperatures have remained at or below 0 ℃ for at least two consecutive years. There is variation in temperature between different years, the bottom of the active layer is not necessarily connected to permafrost table, and a melting sandwich may occur. The author judges the active layer thickness by the change of soil temperature one year. This should be distinguished from the permafrost table.
7. Line 190-193: As I know, there are two alternative schemes for soil organic layer in land surface models, one is assuming one or more organic matter layers cover the mineral layer at a vertical depth, the other is the weighted combination approach, such as in CLM. I suggest that you should compare the two schemes and give their different.
8. Line 343-344, 557-560: I am confused by the description of the lower boundary conditions of the model. The author should clearly state which boundary conditions are used in the model, the Dirichlet condition (fixed temperature in boundary), Neumann conditions (fixed geothermal flow in boundary) or Robin conditions (fixed temperature and geothermal flow in boundary). In addation, the upper boundary conditions should also be properly explained.
9. Line 436-438,455 :You also should give the soil moisture figure using different number of

cycles, and when it stabilizes. Your title is "…a Large Scale Hydrological…", and your results were only soil temperature, how about the soil moisture?

10. Line 466-467: Please check this sentence, the temperature difference reached 1.0 k between 100 times and 2000 times cycles. It revealed that 100 times cycle was not stable, but you said that "there is no significant change after 100 cycles and sometimes less."(In Line 453-454), Why?

11. Line 481-482: The simulations have very longer time period (1979-2016), and the deep soil temperature change was evaluated. As you know, the geothermal flow will have a great influence on the deep ground temperature at a long-time scale, which may be more than the impact of climate change. Strongly recommend that you should use the geothermal flow for the lower boundary by observed data from drilling or the relevant data from references.

12. Line 492-494: It is very confusing here. Active layer thickness is only 3m at JMR. The soil temperature and moisture should be stable values, which are the initial conditions for the next step simulation after 100 cycles (100 years) in theory. However, there were larger differences from simulation results given by Figure 9 because of the initial values of different cycles (50-2000 times). This is very abnormal. You should check the simulation results again, whether the cycle is not enough, or other reasons that make the initial value do not converge. Please give a detailed explanation.

13. Line 527-528: Simulation results of temperature envelopes were lower than observed values, which may be caused by neglecting geothermal flow.

14. Line 554-555: The explanation for the cooling effect of the model increased the depth of SDEP is unreasonable. From Figure 14, it can be seen that the location of SDEP after increasing is located in permafrost, and soil water content in this layer should be frozen throughout the year. I am not sure that the model could take into account the difference in thermal properties between permafrost including ice and ice-free bedrock, and the thermal convection generated by little unfrozen water in the frozen soil. These could explain the cooling effect. If so, further explanations should be provided.

15. Line 575: I suggest that you should check variation in the upper boundary drive (climate) during the simulation time. This may be the reason why the temperature envelope tends to be at a given temperature at lower boundary.

16. The discussion needs be strengthened. You should compare your results with others, then conclude what your new fingdings and contribution.

**Specific comments**

1. Line 101: What is ALD? When you give an abbreviation for the first time, you should give the explain. I found the explain in Line 158, but this is the first time here. In addition, active layer thickness is more commonly used, I suggest use ALT instead of ALD.

2. Line 166: The no (or zero) oscillation depth (ZOD) should be instead of depth of zero annual amplitude (DZAA). DZAA is a professional vocabulary in the field of permafrost research.

---

## Author Comment (AC1) · 20 Sep 2019

**General comments**

Elshamy et al. detail testing and resultant guidelines for the configuration and initialization (especially 'spinning up') of permafrost in large-scale hydrologic models, with a focus in this study of the Mackenzie River Basin. Permafrost exerts primary control on hydrologic routing in cold regions, and thus this topic is critical for Canada and other countries with high latitude regions that are experiencing high rates of warming. As such, the manuscript scope is a good fit for HESS, and the collective authorship team offers many decades of modeling experience and insight. I think the paper will be a useful contribution to HESS, but I think it needs some reworking.

We would like to thank the reviewer for the time spent to carefully review our manuscript. We greatly appreciate the important points raised. We present our response to reviewer's comments below. The reviewer comments are listed below in regular black text, and our response in regular blue text. Some of the reviewer's suggestions have been addressed in the revised manuscript under preparation while other responses point towards what we intend to do further in the manuscript.

**Major concerns**

1. This is a vague concern, but this comes across as a bit of a high-end technical report in places, more than a research paper. Rather than detail why that is, I list a few specific concerns below that map to this general overarching theme.

While we appreciate this comment, this manuscript was written directly as a research paper. Hopefully by addressing the specific concerns below, in addition to the suggested revisions to the manuscript, this concern will be also addressed.

2. This discussion on changing annual discharge is a bit overly simplified. I'd break this down a bit more into different seasons. There is a pretty consistent increase in minimum flows across the pan-Arctic (see, for example, the recent ECCC report, Canada's Changing Climate, or Walvoord and Striegl 2007 GRL, St Jacques and Sauchyn 2009 GRL, Duan et al. 2017 Water – for China)

Thanks for pointing out the importance of seasonal changes to streamflows and for the relevant literature. We intend to revise the discussion on that to reflect the complexity of streamflow response due to differences in seasonal changes based on the suggested literature.

3. The intro is quite long – it is 6 paragraphs, of which several are long. Also, the objectives section which follows is normally embedded in the intro in most papers. This would add about another 2 paragraphs. This needs to be trimmed. Paragraph 4 is especially wordy. Paragraphs 5 and 6 could be cut by 50%.

Thanks for the suggestion to shorten the Introduction & Objectives sections. We intend to revise both sections and remove any unnecessary text to make it more focused.

4. Lists or bulleted sections are not written very parallel in this paper, and they are hard to relate and read. L155-168, L459-467, and L686-691 are examples.

Thanks for the suggestion. We have noted the issue and intend to rephrase the bullets to make them parallel.

5. L205-210, this is very late in the paper to be delineating the focus

The focus was already given in the objectives, especially L122-128. L205-210 were giving the rationale for selecting the sites. However, we are considering shortening Section 3.2 as per your next point and as requested by Reviewer 2. Therefore, the text on L205-210 will be revised as we revise the Introduction and Objectives sections.

6. Because there are three sites, the site description is very long (Sections 3.2.1, 3.2.2, and 3.2.3. This takes up about 7 pages, which is similar in length to short paper on its own. Basically, some of this information (especially the inordinate focus on parameterization, when that is not the point of the study) needs to be moved to an electronic supplement. It detracts from the key messaging, and it's not a very invigorating read. I think the site description is key, but could be shorter, but I don't think the reader needs to wade through endless parameter justification, which could be built into tables in a supplement for interested readers.

Thanks for the suggestions. We agree that Section 3.2 has become too lengthy and we intend to move most of the text into a supplement and keep only relevant parts. This is also suggested by Reviewer 2.

7. Oct. 1979 is certainly late in history as a representative climate from which to base the model spin up. I realize this is briefly addressed later, but I suspect a permafrost modeler would object.

We agree that 1979 may be considered late to start model spin-up for permafrost. However, we discussed that in Section 5 – paragraphs 3-5. Previous work at Norman Wells (Sapriza-Azuri et al., 2018) showed some sensitivity of permafrost conditions to the spin-up year but only if a warm year is selected (Figure 12 in the above mentioned paper). Based on that, the authors suggested to use an average year for the spin-up. We checked that the selected hydrological year (Oct 1979-Sep 1980) is close to an average year based on available records (see L440). We intend to show that in the revised manuscript. There are severe logistical problems in using a longer period. One has to use another climatic dataset to use earlier years as WFDEI only starts in 1979. This means that alternative climatic forcing datasets have to be used and this will have impacts on the results, introducing considerable additional uncertainty. The selected year is performing well for most aspects of our simulation and is resulting in a colder rather than warmer temperatures for the minimum envelopes. The discussion around this point will be strengthened in the revised manuscript.

8. I think a small section on the thermal physics in the model (governing equations, soil freezing curves if any, etc.) would be far more useful to the reader than the emphasis on parameters.

Thanks for the suggestion. We will have that in the revised manuscript.

9. This contribution is very qualitative and even anecdotal in places. For example 'seems' should up 7 times in the manuscript, while 'seem' shows up in 8 places. The difference in model runs are not compared via standard metrics like RMSE or something like that. The discussion seems to rather focus on apparent discrepancies and vague explanations. For examples of this, just consider any section on model comparisons or differences. Also, note recurring appearances of 'much more' and 'too small' – a few actual numbers would be nice.

Thanks for the suggestion. We relied on visual comparisons to assess differences amongst the different simulations. To fully address this comment, we intend to calculate a few standard error metrics to strengthen the visual comparisons by interpolating temperature envelopes at the depths of available observations (which varies from site to site and year to year). In the revised version, we will use some actual numbers, while acknowledging the high uncertainty level in both observations and simulations.

10. The authors do not frame their permafrost modeling results in the discussion around past contributions. Cryosphere scientists have been modeling permafrost and considering spin up scenarios for a very long time. The authors' work is new and interesting (especially the focus on the inclusion of permafrost in large-scale hydrologic modeling), but the thermal physics under consideration are not overly new, and it would make sense to relate their study findings.

Thanks for pointing out this and for describing the work as new and interesting. We intend to revise the discussion to compare the results to other relevant studies.

**Minor concerns**
Many of these are quite trivial

Thanks for helping us improve the manuscript by taking the time to point out these.

L13, comma after 'average' L33, shouldn't basin be capitalized here as elsewhere when preceded by Mackenzie or Mackenzie River?

Changed as advised.

L33, 'heating up by 4 degC' . . ..over what time period? 100,000 years? 50 years?

Revised to: "… by 4°C between 1948 and 2016."

L36, 'American rivers' should just be 'America' and the subsequent semicolon should be a comma

Revised as advised.

L75, 'implied' should be 'inferred'

Revised as advised.

L119, 'In addition to: : :.for spinning' is a fragment

Removed as it was not addressed in the paper.

L143 and elsewhere, 'etc.' occurs in the paper where it is entirely superfluous in a couple places. It tends to look choppy – 'such as' is sufficient.

Removed in L143 and L117 in the revised manuscript.

L141, Land does not need to be capitalized

Capitalization removed.

L152, does this mean that sandstone thermal properties are always used the for the bedrock conductivity everywhere? This seems less than ideal.

Well, we agree it is not ideal, but this is how it is implemented. We have added it to the discussions as a potential improvement to CLASS.

L161 'thus we use a thaw, rather than a freeze criterion' – I have no idea what this means

"The active layer thickness is defined as the thickness of the layer that is subject to annual thawing and freezing in areas underlain by permafrost" (van Everdingen, 2005). "Strictly speaking, the active layer thickness is defined as the lesser of the maximum seasonal frost depth and the maximum seasonal thaw depth" (Walvoord and Kurylyk, 2016). The maximum frost depth can be different from the maximum thaw depth. In case the frost depth is less than the thaw depth, there is a layer above the permafrost that is warmer than $0°C$ but is not connected to the surface (a talik). Because active layer observations are usually based on measuring the maximum thaw depth, we adopted the same criterion when calculating active layer thickness in the model.

L167, This is more commonly called the 'seasonal penetration depth', at least in nonpermafrost regions

We do not disagree with the reviewer on the terminology. However, the work is about permafrost. Therefore, we revised the manuscript to use the more standard term – Depth of Zero Annual Amplitude (DZAA) depth as suggested by Reviewer 2.

L170, permafrost is not defined cryotically like this (frozen vs. unfrozen). It's a temperature definition – i.e. ground below 0C for two or more consecutive years – see, for example, Dobinski, 2011, Earth Science Reviews.

Thanks for pointing this out. We revised the text accordingly and added references.

L173, "MESH/CLASS used to output" should change to 'Prior versions of MESH/CLASS outputted merely temperature profiles' or something like this

Revised to "Prior versions of MESH/CLASS merely outputted …"

175, 'A CLASS typical' should be 'A typical CLASS'

Revised to "A typical CLASS …"

L192, "these has' should be 'these have', and I'm not sure what 'to be carried back to the MRB scale' means

We corrected the phrase to "… these have to be applicable at the MRB scale" to address both parts of the comment. The aim is to establish a methodology that is applicable to the large scale rather than finding the best configuration for the selected sites if it cannot be implemented at the large scale.

L197, 'North West' should be 'Northwest'

Revised as advised.

L216, 'with' should be 'by' and the rest of this sentence needs to be rewritten as it is confusing what it means

We revised the sentence to read: "The basin is located in the sporadic permafrost zone where permafrost underlies few spots only and is characterized by warm temperatures (> -1°C) and limited (<10m) thickness (Smith and Burgess, 2002)".

L220-221, Weather and North should not be capitalized

Revised as advised.

L302, should not be semicolon

Revised as advised.

L304, why does deep permafrost imply no groundwater? It would make more sense to note that the cold climate prevents the formation of a lateral talik, and thus there is no perennial shallow groundwater. See Lamontagne-Halle et al. 2018 (Environmental Research Letters) or Connon et al. 2018, JGR-Earth Surface.

Thanks for pointing out this. We will be revising the text accordingly based on the provided references.

L316 (and elsewhere, do a word search), 'envelops' should be 'envelopes'

Thanks for pointing this out. We checked the manuscript for all instances and corrected accordingly.

**References**
van Everdingen, R. O.: Glossary of Permafrost and Related Ground-Ice Terms., 2005.
Sapriza-Azuri, G., Gamazo, P., Razavi, S. and Wheater, H. S.: On the appropriate definition of soil profile configuration and initial conditions for land surface–hydrology models in cold regions, Hydrol. Earth Syst. Sci., 22(6), 3295–3309, doi:10.5194/hess-22-3295-2018, 2018.
Smith, S. L. and Burgess, M. M.: A digital database of permafrost thickness in Canada., 2002.
Walvoord, M. A. and Kurylyk, B. L.: Hydrologic Impacts of Thawing Permafrost—A Review, Vadose Zo. J., 15(6), 0, doi:10.2136/vzj2016.01.0010, 2016.

---

## Author Comment (AC2) · 20 Sep 2019

This study aims to derive a robust, yet computationally efficient initialization parameterization approach that can be applied to regions where data are scarce and simulations typically require large computational resources. An upscaling approach to inform large-scale ESM simulations based on the insights gained by modelling at small scales was performed. The results show that the model has good performance in reproducing present-climate permafrost properties at the three sites at the Mackenzie River Valley. The results also demonstrate that the simulations are sensitive to the soil layering scheme, the depth to bedrock, and the organic soil properties.

It is really important to investigate the performances of hydrological and land surface models in permafrost regions under climate change. However, there are some shortcomings that might affect the contribution of this study. My main concern and comments are listed as follow.

We would like to thank the reviewer for the time spent to carefully review our manuscript. We greatly appreciate the important points raised. We present our response to reviewer's comments below. The reviewer comments are listed below in regular black text, and our response in regular blue text. Some of the reviewer's suggestions have been addressed in the revised manuscript under preparation while other responses point towards what we intend to do.

**General comments**
1. Lin 34:...however, are not so clear…You should give citations.

The statement is followed by a couple of sentences that provide further explanation and citations and was further strengthened based on comments from Reviewer 1 to read:
"Subsequent impacts on water resources in the region, however, are not well-understood and can be different in different parts. For example, a recent analysis of trends in Arctic freshwater inputs (Durocher et al., 2019) highlights that Eurasian rivers show a significant annual discharge increase during the 1975-2015 period, while in North America, only rivers flowing into the Hudson Bay region in Canada show a significant annual discharge increase during that same period. Canadian rivers flowing directly into the Arctic, of which the Mackenzie River provides the majority of flow, show little change at the annual scale. These analyses at annual scale, however, can mask larger changes at the seasonal scale. For example, Bonsal et al. (2019) report higher winter flows, earlier spring flows, and lower summer flows for some rivers in Canada. Further, they also state that "It is uncertain how projected higher temperatures and reductions in snow cover will combine to affect the frequency and magnitude of future snowmelt-related flooding"".

2. Line 39: What do you mean "uncertainty"?

With the modifications to the first paragraph given above, we rephrased our statement to read:

"The hydrological response of cold regions to climate change is highly uncertain with the current state of knowledge, because, to a large extent, of our limited understanding and representation of how the different hydrologic processes will interact under new climate conditions".

The introduction will be refocused further in the revised manuscript.

3. Line 46-50: You give importance of permafrost here, which may be not suite for this paragraph. I suggest that you provide separate paragraph to show the importance of permafrost and the progress in interaction between permafrost and hydrological at the beginning of the introduction.

As we intend to refocus the introduction, this will be addressed in the revised manuscript.

4. Line 51 and 91: Here the authors give the modeling work in hydrological processes in permafrost regions, I noticed that the models were all land surface models. As I know, there were many modeling work that has been done by hydrological models in cold regions, such as VIC, GBHM. I would suggest that the authors to provide the different with hydrological models and land surface models on the previous modeling work in hydrological processes in cold regions, then clearly state why you choose the land surface model for this study.

While the contributions of the mentioned studies are significant, the emphasis herein was to consider those models that include robust representation of the energy balance and are able to produce detailed temperature profiles in multi-layer deep soil columns. Generally, hydrological models do not include the full energy balance and therefore they do not have a handle on permafrost unless they are coupled with other energy balance models, as Zhang et al. (2012) did with GBHM. VIC (Liang et al., 1994) is a special case of hydrological models and is often described as a land surface hydrological model which makes it similar to MESH in this regard. The modelling efforts also include thermal modelling (e.g. Wright et al., 2003). We intend to revise the introduction to make it more focused and to add some references to reviews of permafrost modelling such as Riseborough et al. (2008) and Walvoord and Kurylyk (2016) to guide the interested reader.

5. Section 3.2 Study Sites and Data: This section is too long. Please make it concise using figures and tables. In addition, you may combine Section 3.5 (Climate Forcing) with this Section. They are all data introduction.

We agree with the Reviewer that this section is too long and we intend to shorten it and move most of the details to a supplement. This is also suggested by Reviewer 1.

6. Line 170-171, Permafrost, which is defined as ground in which temperatures have remained at or below 0°C for at least two consecutive years. There is variation in temperature between different years, the bottom of the active layer is not necessarily connected to permafrost table, and a melting sandwich may occur. The author judges the active layer thickness by the change of soil temperature one year. This should be distinguished from the permafrost table.

Thanks for pointing up this discussion. We fully agree with the reviewer and that is the reason we use a "thaw rather than freeze criterion" in the definition of the ALT (lines 158-162) and explicitly mention that it has to be connected to the surface. We will be revising the text to emphasize this difference in the revised manuscript.

7. Line 190-193: As I know, there are two alternative schemes for soil organic layer in land surface models, one is assuming one or more organic matter layers cover the mineral layer at a vertical depth, the other is the weighted combination approach, such as in CLM. I suggest that you should compare the two schemes and give their different.

CLASS can either use a percentage of organic matter within a mineral soil layer or use fully organic layers. In the first case, the organic content is used to modify soil hydraulic and thermal properties - similar to CLM (Oleson et al., 2013). In the latter, CLASS has special values for those properties depending on the type of organic soil selected (Fibric, Hemic or Sapric) based on the work of Letts et

al. (2000) for peat soils. This is described in the manuscript in L184-193. We adopted a 30% threshold to differentiate fully organic soils from mineral soils with organic matter based on Soil Classification Working Group recommendations.

L367-378: For the HPC site, we tested both approaches as the organic matter was only 18% (below the 30% threshold). We selected to use fully organic soils for BWC and JMR sites because of the high percentage of organic matter found from the soil dataset we used (above the 30% threshold). We thought a mineral soil would not be suitable for those sites. However, we have also conducted simulations at all sites using the mineral soil formulation with high organic percentage for the BWC and JMR sites and intend to discuss the differences in soil properties and their impact on the simulations in the revised manuscript.

8. Line 343-344, 557-560: I am confused by the description of the lower boundary conditions of the model. The author should clearly state which boundary conditions are used in the model, the Dirichlet condition (fixed temperature in boundary), Neumann conditions (fixed geothermal flow in boundary) or Robin conditions (fixed temperature and geothermal flow in boundary). In addation, the upper boundary conditions should also be properly explained.

CLASS uses a constant geothermal flux at the bottom boundary (i.e. Neumann type condition – constant derivative). We used the default value for this flux (zero) and thus used the term no-flux boundary as mentioned in L343-344 and on L559. We will revise the manuscript to further clarify that. We noticed in simulations with shallower soil column depths that the temperature at the bottom boundary changes over time as mentioned in L461-463, which confirms that the boundary condition is not type 1 or 3 (Dirichlet or Robin). The Upper boundary condition depends on the meteorological forcing and how it is modified by the canopy and snow cover to determine the heat flux at the soil surface. Following the recommendations of Reviewer 1, we intend to add a section on the mathematical formulation in the revised manuscript that should clarify the matter.

9. Line 436-438、455 :You also should give the soil moisture figure using different number of cycles, and when it stabilizes. Your title is "…a Large Scale Hydrological…", and your results were only soil temperature, how about the soil moisture?

We agree with the reviewer and we intend to add figures of soil moisture profiles and convergence for a few cases to illustrate the point.

10. Line 466-467: Please check this sentence, the temperature difference reached 1.0 k between 100 times and 2000 times cycles. It revealed that 100 times cycle was not stable, but you said that "there is no significant change after 100 cycles and sometimes less."(In Line 453-454), Why?

Thanks for pointing out this potential contradiction. We think that a temperature change of 1K over a period of 1900 years (cycles) is negligible. That's about 0.0005 K/year (cycle). This is not visible in Figure 7 where we plot the temperature profiles but is more visible on Figure 8 where the temperature sequence is plotted. We amended the statement on L466-467 to include the rate to emphasize that in the revised manuscript under preparation. Additionally, the impact of the number of spinning cycles on the simulation of ALT and temperature envelopes is shown to be minimal in section 4.2.

11. Line 481-482: The simulations have very longer time period (1979-2016), and the deep soil temperature change was evaluated. As you know, the geothermal flow will have a great influence on the deep ground temperature at a long-time scale, which may be more than the impact of climate change. Strongly recommend that you should use the geothermal flow for the lower boundary by observed data from drilling or the relevant data from references.

We have done additional simulations using geothermal heat flux and will be reporting on that in the revised manuscript. They basically emphasize the previous findings of Sapriza-Azuri et al. (2018) for Norman Wells using the same land surface model we used (CLASS) that the geothermal flux has negligible impact on the results. In this paper, the authors compared two scenarios: 1) no heat flow at the bottom of the lowest soil layer, 2) a constant geothermal flow of 0.083 Wm-2 based on local measurement in Normal Wells. The scenarios were applied for a climate average year spin-up by 2000 cycles to several soil depth configurations and parameter values. Results reported by authors showed, as stated in the manuscript L342-347, that the impact of geothermal flux was minimal and the temperature difference between the two scenarios was small in most simulations and is within ±0.15°C in approximately 60 % of simulations. In fact, 1979-2016 is quite a short period specially to catch big difference for the deep soil temperature.  In that sense Sapriza-Azuri et al. (2018) used a 2000-year simulation without getting too much difference.

12. Line 492-494: It is very confusing here. Active layer thickness is only 3m at JMR. The soil temperature and moisture should be stable values, which are the initial conditions for the next step simulation after 100 cycles (100 years) in theory. However, there were larger differences from simulation results given by Figure 9 because of the initial values of different cycles (50-2000 times). This is very abnormal. You should check the simulation results again, whether the cycle is not enough, or other reasons that make the initial value do not converge. Please give a detailed explanation.

The less stable conditions at JMR are possibly related to the small thickness of permafrost and the thick organic layers. These may have caused the drifting in temperature shown in Figure 7 for some layers under the slightly warmer conditions compared to HPC and BWC. We intend to check the results again to find a better explanation for the phenomenon and clarify that in the revised manuscript.

13. Line 527-528: Simulation results of temperature envelopes were lower than observed values, which may be caused by neglecting geothermal flow.

As mentioned in our response to point #11 above, we conducted simulations for both JMR and BWC using the geothermal flux and it had minor effect as we will be reporting. We are investigating the reasons, which might be related to the quality of snow simulations as well as the configuration of organic soils and the parameter values of the soil (drainage and thermal) set for such places.

14. Line 554-555: The explanation for the cooling effect of the model increased the depth of SDEP is unreasonable. From Figure 14, it can be seen that the location of SDEP after increasing is located in permafrost, and soil water content in this layer should be frozen throughout the year. I am not sure that the model could take into account the difference in thermal properties between permafrost including ice and ice-free bedrock, and the thermal convection generated by little unfrozen water in the frozen soil. These could explain the cooling effect. If so, further explanations should be provided.

We agree with the reviewer. SDEP remains below the active layer and therefore any moisture will be frozen. CLASS differentiates between ice-free bedrock (below SDEP) and permafrost that contains ice. However, we intend to further investigate the soil moisture content and to compare the thermal properties of the soil above and below SDEP to see if the differences can explain the cooling. Thanks for the ideas to close this loose end.

15. Line 575: I suggest that you should check variation in the upper boundary drive (climate) during the simulation time. This may be the reason why the temperature envelope tends to be at a given temperature at lower boundary.

The upper boundary condition (climate) is transient for the 1979-2016 simulation, yet the temperature of the lowest layer barely changes over that period. We tested with shallower soil profiles and found it more responsive to changes. We think that the thermal properties and deep profile are the reasons for having such response at the lower boundary. We intend to analyze the forcing climate as well as the thermal soil properties to further address this concern and revise the manuscript accordingly.

16. The discussion needs be strengthened. You should compare your results with others, then conclude what your new fingdings and contribution.

Thanks for pointing this out. We intend to strengthen the discussions in the revised manuscript by framing it around previous work to better show the contribution. This is also suggested by Reviewer 1.

**Specific comments**
1. Line 101: What is ALD? When you give an abbreviation for the first time, you should give the explain. I found the explain in Line 158, but this is the first time here. In addition, active layer thickness is more commonly used, I suggest use ALT instead of ALD.

Thanks for noting this. ALD and ALT are equivalent because our model does not include land settlement and therefore the fixed reference level used to measure ALD is the ground surface - definition is given in Geological Survey Canada reports (e.g. Smith et al., 2004). However, we changed ALD to ALT in the whole document (inducing figures) to use the more standard terminology. We made sure all terms are spelled out on first use.

2. Line 166: The no (or zero) oscillation depth (ZOD) should be instead of depth of zero annual amplitude (DZAA). DZAA is a professional vocabulary in the field of permafrost research.

As we replaced ALD with ALT, we replaced ZOD with DZAA in the whole document to be using the standard terminology of permafrost research.

**References**
Bonsal, B. R., Peters, D. L., Seglenieks, F., Rivera, A. and Berg, A.: Changes in freshwater availability across Canada, in Canada's Changing Climate Report, pp. 261–342. [online] Available from: https://www.nrcan.gc.ca/sites/www.nrcan.gc.ca/files/energy/Climate-change/pdf/CCCR-Chapter6-ChangesInFreshwaterAvailabilityAcrossCanada.pdf (Accessed 27 August 2019), 2019.
Durocher, M., Requena, A. I., Burn, D. H. and Pellerin, J.: Analysis of trends in annual streamflow to the Arctic Ocean, Hydrol. Process., 33(7), 1143–1151, doi:10.1002/hyp.13392, 2019.
Letts, M. G., Roulet, N. T., Comer, N. T., Skarupa, M. R. and Verseghy, D. L.: Parameterization of Peatland Hydraulic Properties for the Canadian Land Surface Scheme, ATMOSPHERE-OCEAN, 38(1), doi:10.1080/07055900.2000.9649643, 2000.
Liang, X., Lettenmaier, D. P., Wood, E. F. and Burges, S. J.: A simple hydrologically based model of land surface water and energy fluxes for general circulation models, J. Geophys. Res., 99(D7), 14415, doi:10.1029/94JD00483, 1994.
Oleson, K. W., Lawrence, D. M., Bonan, G. B., Drewniak, B., Huang, M., Charles, D., Levis, S., Li, F., Riley, W. J., Zachary, M., Swenson, S. C., Thornton, P. E., Bozbiyik, A., Fisher, R., Heald, C. L., Kluzek, E., Lamarque, F., Lawrence, P. J., Leung, L. R., Muszala, S., Ricciuto, D. M. and Sacks, W.: Technical Description of version 4.5 of the Community Land Model (CLM) Coordinating., 2013.
Riseborough, D., Shiklomanov, N., Etzelmüller, B., Gruber, S. and Marchenko, S.: Recent advances in permafrost modelling, Permafr. Periglac. Process., 19(2), 137–156, doi:10.1002/ppp.615, 2008.
Sapriza-Azuri, G., Gamazo, P., Razavi, S. and Wheater, H. S.: On the appropriate definition of soil profile configuration and initial conditions for land surface–hydrology models in cold regions, Hydrol. Earth Syst. Sci., 22(6), 3295–3309, doi:10.5194/hess-22-3295-2018, 2018.

Smith, S. L., Burgess, M. M., Riseborough, D., Coultish, T. and Chartrand, J.: Digital summary database of permafrost and thermal conditions - Norman Wells pipeline study sites, Geol. Surv. Canada, Open File 4635, 4635, 1–104, doi:10.4095/215482, 2004.

Walvoord, M. A. and Kurylyk, B. L.: Hydrologic Impacts of Thawing Permafrost—A Review, Vadose Zo. J., 15(6), 0, doi:10.2136/vzj2016.01.0010, 2016.

Wright, J. F., Duchesne, C. and Côté, M. M.: Regional-scale permafrost mapping using the TTOP ground temperature model, in Proceedings 8th International Conference on Permafrost, pp. 1241–1246. [online] Available from: http://research.iarc.uaf.edu/NICOP/DVD/ICOP 2003 Permafrost/Pdf/Chapter_218.pdf (Accessed 19 April 2019), 2003.

Zhang, Y., Cheng, G., Li, X., Han, X., Wang, L., Li, H., Chang, X. and Flerchinger, G. N.: Coupling of a simultaneous heat and water model with a distributed hydrological model and evaluation of the combined model in a cold region watershed, , doi:10.1002/hyp.9514, 2012.

---

## Author Response (AR1)

**General comments**

Elshamy et al. detail testing and resultant guidelines for the configuration and initialization (especially 'spinning up') of permafrost in large-scale hydrologic models, with a focus in this study of the Mackenzie River Basin. Permafrost exerts primary control on hydrologic routing in cold regions, and thus this topic is critical for Canada and other countries with high latitude regions that are experiencing high rates of warming. As such, the manuscript scope is a good fit for HESS, and the collective authorship team offers many decades of modeling experience and insight. I think the paper will be a useful contribution to HESS, but I think it needs some reworking.

We would like to thank the reviewer for the time spent to carefully review our manuscript. We greatly appreciate the important points raised. We present our response to reviewer's comments below. The reviewer comments are listed below in regular black text, and our response in regular blue text. Some of the reviewer's suggestions have been addressed in the revised manuscript under preparation while other responses point towards what we intend to do further in the manuscript.

**Major concerns**

1. This is a vague concern, but this comes across as a bit of a high-end technical report in places, more than a research paper. Rather than detail why that is, I list a few specific concerns below that map to this general overarching theme.

While we appreciate this comment, this manuscript was written directly as a research paper. Hopefully by addressing the specific concerns below, in addition to the various revisions made to the manuscript, this concern will have been also addressed.

2. This discussion on changing annual discharge is a bit overly simplified. I'd break this down a bit more into different seasons. There is a pretty consistent increase in minimum flows across the pan-Arctic (see, for example, the recent ECCC report, Canada's Changing Climate, or Walvoord and Striegl 2007 GRL, St Jacques and Sauchyn 2009 GRL, Duan et al. 2017 Water – for China)

Thanks for pointing out the importance of seasonal changes to streamflows and for the relevant literature. The discussion on that has been revised to reflect the complexity of streamflow response due to differences in seasonal changes based on the suggested literature – see L39-45 in the revised manuscript.

3. The intro is quite long – it is 6 paragraphs, of which several are long. Also, the objectives section which follows is normally embedded in the intro in most papers. This would add about another 2 paragraphs. This needs to be trimmed. Paragraph 4 is especially wordy. Paragraphs 5 and 6 could be cut by 50%.

Thanks for the suggestion to shorten the Introduction & Objectives sections. We have refocused the Introduction and shortened the paragraphs. We also removed the Objectives section and added a short paragraph for it at the end of the Introduction. Please see the revised Introduction Section.

4. Lists or bulleted sections are not written very parallel in this paper, and they are hard to relate and read. L155-168, L459-467, and L686-691 are examples.

Thanks for the suggestion. We have noted the issue and rephrased the bullets throughout the document to make them parallel.

5. L205-210, this is very late in the paper to be delineating the focus

The focus was already given in the objectives, now in L124-130, integrating the rationale for selecting the sites that was given by L205-210 in the original manuscript.

6. Because there are three sites, the site description is very long (Sections 3.2.1, 3.2.2, and 3.2.3. This takes up about 7 pages, which is similar in length to short paper on its own. Basically, some of this information (especially the inordinate focus on parameterization, when that is not the point of the study) needs to be moved to an electronic supplement. It detracts from the key messaging, and it's not a very invigorating read. I think the site description is key, but could be shorter, but I don't think the reader needs to wade through endless parameter justification, which could be built into tables in a supplement for interested readers.

Thanks for the suggestions. This is also suggested by Reviewer 2. We agree that Section 3.2 has become too lengthy and moved most of the text into a supplement and kept only relevant parts, in Section 2.2 of the revised manuscript.

7. Oct. 1979 is certainly late in history as a representative climate from which to base the model spin up. I realize this is briefly addressed later, but I suspect a permafrost modeler would object.

We agree that 1979 may be considered late to start model spin-up for permafrost. Previous work at Norman Wells (Sapriza-Azuri et al., 2018) showed some sensitivity of permafrost conditions to the spinup year but only if a warm year is selected (Figure 12 in the above mentioned paper). Based on that, the authors suggested to use an average year for the spin-up. We checked that the selected hydrological year (Oct 1979-Sep 1980) is close to an average year based on available records (see Table 4 in the revised manuscript). There are severe logistical problems in using a longer period. One has to use another climatic dataset to use earlier years as WFDEI only starts in 1979. This means that alternative climatic forcing datasets have to be used and this will have impacts on the results, introducing considerable additional uncertainty. The selected year is performing well for most aspects of our simulation and is resulting in a colder rather than warmer temperatures for the minimum envelopes. Section 2.4 in revised manuscript gives some detail about the selection of the climate forcing dataset.

8. I think a small section on the thermal physics in the model (governing equations, soil freezing curves if any, etc.) would be far more useful to the reader than the emphasis on parameters.

Thanks for the suggestion. We have added that to Section 2.1 of the revised manuscript with more details in Section S1 of the supplementary material.

9. This contribution is very qualitative and even anecdotal in places. For example 'seems' should up 7 times in the manuscript, while 'seem' shows up in 8 places. The difference in model runs are not compared via standard metrics like RMSE or something like that. The discussion seems to rather focus on apparent discrepancies and vague explanations. For examples of this, just consider any section on model comparisons or differences. Also, note recurring appearances of 'much more' and 'too small' – a few actual numbers would be nice.

Thanks for the suggestion. We relied on visual comparisons to assess differences amongst the different simulations. To fully address this comment, we calculated RMSE for ALT, DZAA, and temperature envelopes in all results sections of the revised version. Despite the high uncertainty level in both observations and simulations, we have tried to use more definitive language and logical explanations.

10. The authors do not frame their permafrost modeling results in the discussion around past contributions. Cryosphere scientists have been modeling permafrost and considering spin up scenarios for a very long time. The authors' work is new and interesting (especially the focus on the inclusion of permafrost in large-scale hydrologic modeling), but the thermal physics under consideration are not overly new, and it would make sense to relate their study findings.

Thanks for pointing out this and for describing the work as new and interesting. We revised the discussion to compare the results to other relevant studies.

**Minor concerns**

Many of these are quite trivial

Thanks for helping us improve the manuscript by taking the time to point out these.

L13, comma after 'average' L33, shouldn't basin be capitalized here as elsewhere when preceded by Mackenzie or Mackenzie River?

**Changed as advised.**

L33, 'heating up by 4 degC' . . . . over what time period? 100,000 years? 50 years?

Revised to: "... by 4°C between 1948 and 2016."

L36, 'American rivers' should just be 'America' and the subsequent semicolon should be a comma

**Revised as advised.**

L75, 'implied' should be 'inferred'

Revised as advised.

L119, 'In addition to: : :.for spinning' is a fragment

Removed as it was not addressed in the paper.

L143 and elsewhere, 'etc.' occurs in the paper where it is entirely superfluous in a couple places. It tends to look choppy – 'such as' is sufficient.

Removed in the revised manuscript.

L141, Land does not need to be capitalized

Capitalization removed.

L152, does this mean that sandstone thermal properties are always used the for the bedrock conductivity everywhere? This seems less than ideal.

Well, we agree it is not ideal, but this is how it is implemented. We have added it to the discussions as a potential improvement to CLASS.

L161 'thus we use a thaw, rather than a freeze criterion' – I have no idea what this means

"The active layer thickness is defined as the thickness of the layer that is subject to annual thawing and freezing in areas underlain by permafrost" (van Everdingen, 2005). "Strictly speaking, the active layer thickness is defined as the lesser of the maximum seasonal frost depth and the maximum seasonal thaw depth" (Walvoord and Kurylyk, 2016). The maximum frost depth can be different from the maximum thaw depth. In case the frost depth is less than the thaw depth, there is a layer above the permafrost that is warmer than 0°C but is not connected to the surface (a talik). Because active layer observations are usually based on measuring the maximum thaw depth, we adopted the same criterion when calculating active layer thickness in the model. This has been clarified further in L169-179 of the revised manuscript.

L167, This is more commonly called the 'seasonal penetration depth', at least in nonpermafrost regions

We do not disagree with the reviewer on the terminology. However, the work is about permafrost. Therefore, we revised the manuscript to use the more standard term – Depth of Zero Annual Amplitude (DZAA) depth as suggested by Reviewer 2.

L170, permafrost is not defined cryotically like this (frozen vs. unfrozen). It's a temperature definition – i.e. ground below 0C for two or more consecutive years – see, for example, Dobinski, 2011, Earth Science Reviews.

Thanks for pointing this out. We revised the text accordingly and added references. See L169-179 of the revised manuscript.

L173, "MESH/CLASS used to output" should change to 'Prior versions of MESH/CLASS outputted merely temperature profiles' or something like this

Revised to "Prior versions of MESH/CLASS merely outputted ..." on L180 of the revised manuscript.

175, 'A CLASS typical' should be 'A typical CLASS'

Revised to "A typical CLASS ..." on L181 of the revised manuscript.

L192, "these has' should be 'these have', and I'm not sure what 'to be carried back to the MRB scale' means

The aim is to establish a methodology that is applicable to the large scale rather than finding the best configuration for the selected sites if it cannot be implemented at the large scale. This has been included on L298-299 of the revised manuscript.

L197, 'North West' should be 'Northwest'

**Revised as advised.**

L216, 'with' should be 'by' and the rest of this sentence needs to be rewritten as it is confusing what it means

We revised the sentence to read: "The basin is located in the sporadic permafrost zone where permafrost underlies few spots only and is characterized by warm temperatures (> -1°C) and limited (<10m) thickness (Smith and Burgess, 2002)". Now on L250-252 of the revised manuscript.

L220-221, Weather and North should not be capitalized

Revised as advised.

L302, should not be semicolon

Revised as advised – Now in supplementary material.

L304, why does deep permafrost imply no groundwater? It would make more sense to note that the cold climate prevents the formation of a lateral talik, and thus there is no perennial shallow groundwater. See Lamontagne-Halle et al. 2018 (Environmental Research Letters) or Connon et al. 2018, JGR-Earth Surface.

Thanks for pointing out this. We revised the text, which was moved to the supplementary material.

L316 (and elsewhere, do a word search), 'envelops' should be 'envelopes'

Thanks for pointing this out. We checked the manuscript for all instances and corrected accordingly.

**References**

van Everdingen, R. O.: Glossary of Permafrost and Related Ground-Ice Terms., 2005. Sapriza-Azuri, G., Gamazo, P., Razavi, S. and Wheater, H. S.: On the appropriate definition of soil profile configuration and initial conditions for land surface–hydrology models in cold regions, Hydrol. Earth Syst. Sci., 22(6), 3295–3309, doi:10.5194/hess-22-3295-2018, 2018. Smith, S. L. and Burgess, M. M.: A digital database of permafrost thickness in Canada., 2002.

Walvoord, M. A. and Kurylyk, B. L.: Hydrologic Impacts of Thawing Permafrost—A Review, Vadose Zo. J., 15(6), 0, doi:10.2136/vzj2016.01.0010, 2016.

This study aims to derive a robust, yet computationally efficient initialization parameterization approach that can be applied to regions where data are scarce and simulations typically require large computational resources. An upscaling approach to inform large-scale ESM simulations based on the insights gained by modelling at small scales was performed. The results show that the model has good performance in reproducing present-climate permafrost properties at the three sites at the Mackenzie River Valley. The results also demonstrate that the simulations are sensitive to the soil layering scheme, the depth to bedrock, and the organic soil properties.

It is really important to investigate the performances of hydrological and land surface models in permafrost regions under climate change. However, there are some shortcomings that might affect the contribution of this study. My main concern and comments are listed as follow.

We would like to thank the reviewer for the time spent to carefully review our manuscript. We greatly appreciate the important points raised. We present our response to reviewer's comments below. The reviewer comments are listed below in regular black text, and our response in regular blue text. Some of the reviewer's suggestions have been addressed in the revised manuscript under preparation while other responses point towards what we intend to do.

**General comments**

1. Lin 34:...however, are not so clear...You should give citations.

The statement is followed by a couple of sentences that provide further explanation and citations and was further strengthened based on comments from Reviewer 1. Please check L34-45 of the revised manuscript.

2. Line 39: What do you mean "uncertainty"?

With the modifications to the first paragraph given above, we rephrased our statement to read:

"The hydrological response of cold regions to climate change is highly uncertain ...", now on L50-53 of the revised manuscript. The introduction have be refocused in the revised manuscript.

3. Line 46-50: You give importance of permafrost here, which may be not suite for this paragraph. I suggest that you provide separate paragraph to show the importance of permafrost and the progress in interaction between permafrost and hydrological at the beginning of the introduction.

As we have refocused the introduction, this has been addressed in paragraph 2 of the Introduction in the revised manuscript.

4. Line 51 and 91: Here the authors give the modeling work in hydrological processes in permafrost regions, I noticed that the models were all land surface models. As I know, there were many modeling work that has been done by hydrological models in cold regions, such as VIC, GBHM. I

would suggest that the authors to provide the different with hydrological models and land surface models on the previous modeling work in hydrological processes in cold regions, then clearly state why you choose the land surface model for this study.

While the contributions of the mentioned studies are significant, the emphasis herein was to consider those models that include robust representation of the energy balance and are able to produce detailed temperature profiles in multi-layer deep soil columns. Generally, hydrological models do not include the full energy balance and therefore they do not have a handle on permafrost unless they are coupled with other energy balance models, as Zhang et al. (2012) did with GBHM. VIC (Liang et al., 1994) is a special case of hydrological models and is often described as a land surface hydrological model which makes it similar to MESH in this regard. The modelling efforts also include thermal modelling (e.g. Wright et al., 2003) as mentioned in the manuscript. We revised the introduction and added some references to reviews of permafrost modelling such as Riseborough et al. (2008) and Walvoord and Kurylyk (2016) to guide the interested reader. See L59-62 of the revised manuscript.

5. Section 3.2 Study Sites and Data: This section is too long. Please make it concise using figures and tables. In addition, you may combine Section 3.5 (Climate Forcing) with this Section. They are all data introduction.

We agree with the Reviewer that Section 3.2 is too long (2.2 in the revised manuscript) and we have shortened it and moved most of the details to a supplement. This is also suggested by Reviewer 1. However, we kept the Climate Forcing (now Section 2.4) as a separate section.

6. Line 170-171, Permafrost, which is defined as ground in which temperatures have remained at or below 0°C for at least two consecutive years. There is variation in temperature between different years, the bottom of the active layer is not necessarily connected to permafrost table, and a melting sandwich may occur. The author judges the active layer thickness by the change of soil temperature one year. This should be distinguished from the permafrost table.

Thanks for pointing up this discussion. We fully agree with the reviewer and that is the reason we use a "thaw rather than freeze criterion" in the definition of the ALT and explicitly mention that it has to be connected to the surface. We revised the text to emphasize this difference. Please see L169-179 in the revised manuscript.

7. Line 190-193: As I know, there are two alternative schemes for soil organic layer in land surface models, one is assuming one or more organic matter layers cover the mineral layer at a vertical depth, the other is the weighted combination approach, such as in CLM. I suggest that you should compare the two schemes and give their different.

CLASS can either use a percentage of organic matter within a mineral soil layer or use fully organic layers. In the first case, the organic content is used to modify soil hydraulic and thermal properties - similar to CLM (Oleson et al., 2013). In the latter, CLASS has special values for those properties depending on the type of organic soil selected (Fibric, Hemic or Sapric) based on the work of Letts et al. (2000) for peat soils. This has been clarified in the revised manuscript in L213-228. We conducted additional simulations using the two alternative ways for all three sites and compared them in the revised manuscript Sections 3.1 and 3.2.

8. Line 343-344, 557-560: I am confused by the description of the lower boundary conditions of the model. The author should clearly state which boundary conditions are used in the model, the Dirichlet condition (fixed temperature in boundary), Neumann conditions (fixed geothermal flow in boundary) or Robin conditions (fixed temperature and geothermal flow in boundary). In addation, the upper boundary conditions should also be properly explained.

CLASS uses a constant geothermal flux at the bottom boundary (i.e. Neumann type condition – constant derivative). We used the default value for this flux (zero) and thus used the term no-flux boundary as mentioned in L343-344 and on L559 of the original manuscript. We noticed in simulations with shallower soil column depths that the temperature at the bottom boundary changes over time as mentioned in L461-463, which confirms that the boundary condition is not type 1 or 3 (Dirichlet or Robin). The Upper boundary condition depends on the meteorological forcing and how it is modified by the canopy and snow cover to determine the heat flux at the soil surface. Following the recommendations of Reviewer 1, we extended Section 2.1 to include the mathematical formulation with more details in Section S1 of the supplementary material.

9. Line 436-438, 455 :You also should give the soil moisture figure using different number of cycles, and when it stabilizes. Your title is "...a Large Scale Hydrological...", and your results were only soil temperature, how about the soil moisture?

We agree with the reviewer and we have added figures of soil moisture profiles and convergence for a few cases to illustrate the point – see Figure 6 and Section 3.1 of the revised manuscript.

10. Line 466-467: Please check this sentence, the temperature difference reached 1.0 k between 100 times and 2000 times cycles. It revealed that 100 times cycle was not stable, but you said that "there is no significant change after 100 cycles and sometimes less." (In Line 453-454), Why?

Thanks for pointing out this potential contradiction. We think that a temperature change of 1K over a period of 1900 years (cycles) is negligible. That's about 0.0005 K/year (cycle). This was not visible in Figure 7 (Figure 5 in the revised manuscript) where we plot the temperature profiles but is more visible on Figure 8 (Figure 7 in the revised manuscript) where the temperature sequence is plotted. We revised the manuscript to explain why such drifts occur in some cases. Additionally, the impact of the number of spinning cycles on the simulation of ALT and temperature envelopes is shown to be minimal in section 3.2 of the revised manuscript.

11. Line 481-482: The simulations have very longer time period (1979-2016), and the deep soil temperature change was evaluated. As you know, the geothermal flow will have a great influence on the deep ground temperature at a long-time scale, which may be more than the impact of climate change. Strongly recommend that you should use the geothermal flow for the lower boundary by observed data from drilling or the relevant data from references.

We have done additional simulations using geothermal heat flux and reported on that in the revised manuscript. They basically emphasize the previous findings of Sapriza-Azuri et al. (2018) for Norman Wells using the same land surface model we used (CLASS) that the geothermal flux has negligible impact on the results. In there paper, the authors compared two scenarios: 1) no heat flow at the bottom of the lowest soil layer, 2) a constant geothermal flow of 0.083 Wm-2 based on local measurement in Normal Wells. The scenarios were applied for a climate average year spin-up by 2000 cycles to several soil depth configurations and parameter values. Results reported by authors showed, as stated in the revised manuscript Section 3.4, that the impact of geothermal flux was minimal and the temperature difference between the two scenarios was small in most simulations and is within ±0.15°C in approximately 60 % of simulations. In fact, 1979-2016 is quite a short period specially to catch big differences for the deep soil temperature. In that sense Sapriza-Azuri et al. (2018) used a 2000-year simulation without getting too much difference. Our simulations confirmed those findings.

12. Line 492-494: It is very confusing here. Active layer thickness is only 3m at JMR. The soil temperature and moisture should be stable values, which are the initial conditions for the next step

simulation after 100 cycles (100 years) in theory. However, there were larger differences from simulation results given by Figure 9 because of the initial values of different cycles (50-2000 times). This is very abnormal. You should check the simulation results again, whether the cycle is not enough, or other reasons that make the initial value do not converge. Please give a detailed explanation.

We checked the results and did further investigations. We found that the less stable conditions at JMR are related to the thick organic layers. We also found that the water and ice contents to play a role in such situations. These have also caused the drifting in temperature shown for some layers in Figure 5 of the revised manuscript under the slightly warmer conditions at JMR compared to HPC and BWC. The explanation is discussed in the revised manuscript L437-453.

13. Line 527-528: Simulation results of temperature envelopes were lower than observed values, which may be caused by neglecting geothermal flow.

As mentioned in our response to point #11 above, we conducted simulations for all sites using the geothermal flux and it had minor effect as we will be reporting. We investigated the reasons of simulating colder than observed temperatures and found it to be related to the configuration of organic soils and the parameter values of the soil. The colder winter temperatures near the surface under peat soils was reported in the literature as discussed in the revised manuscript (see L507-512) and in Section 3.5. The quality of snow simulations cannot be over-ruled but it is beyond the scope of the paper to assess that.

14. Line 554-555: The explanation for the cooling effect of the model increased the depth of SDEP is unreasonable. From Figure 14, it can be seen that the location of SDEP after increasing is located in permafrost, and soil water content in this layer should be frozen throughout the year. I am not sure that the model could take into account the difference in thermal properties between permafrost including ice and ice-free bedrock, and the thermal convection generated by little unfrozen water in the frozen soil. These could explain the cooling effect. If so, further explanations should be provided.

We agree with the reviewer. SDEP remains below the active layer and therefore any moisture in between will be frozen. CLASS differentiates between ice-free bedrock (below SDEP) and permafrost that contains ice. However, we further investigated the soil moisture content and checked the thermal properties of the soil above and below SDEP and expressed our findings on L547-550 of the revised manuscript.

15. Line 575: I suggest that you should check variation in the upper boundary drive (climate) during the simulation time. This may be the reason why the temperature envelope tends to be at a given temperature at lower boundary.

The upper boundary condition (climate) is transient for the 1979-2016 simulation, yet the temperature of the lowest layer barely changes over that period. We tested with shallower soil profiles and found it more responsive to changes. We beleive that the thermal properties and deep profile are the reasons for having such response at the lower boundary. Observations show small changes over the same period at the deepest observational levels which are not more than 15m (see Figure S1 in the supplemtary material) which gives reason to believe that changes at 50m would be negligible and that the model is behaving normally.

16. The discussion needs be strengthened. You should compare your results with others, then conclude what your new fingdings and contribution.

Thanks for pointing this out. We strengthened the discussions in the revised manuscript by framing it around previous work to better show the contribution as also suggested by Reviewer 1.

**Specific comments**

1. Line 101: What is ALD? When you give an abbreviation for the first time, you should give the explain. I found the explain in Line 158, but this is the first time here. In addition, active layer thickness is more commonly used, I suggest use ALT instead of ALD.

Thanks for noting this. ALD and ALT are equivalent because our model does not include land settlement and therefore the fixed reference level used to measure ALD is the ground surface - definition is given in Geological Survey Canada reports (e.g. Smith et al., 2004). However, we changed ALD to ALT in the whole document (inducing figures) to use the more standard terminology. We made sure all terms are spelled out on first use.

2. Line 166: The no (or zero) oscillation depth (ZOD) should be instead of depth of zero annual amplitude (DZAA). DZAA is a professional vocabulary in the field of permafrost research.

As we replaced ALD with ALT, we replaced ZOD with DZAA in the whole document to be using the standard terminology of permafrost research.

**On the Configuration and Initialization of a Large Scale Hydrological Land Surface**

**Model to Represent Permafrost**

Mohamed E. Elshamy1\*, Daniel Princz2, Gonzalo Sapriza-Azuri3, Mohamed S. Abdelhamed1, Al Pietroniro1,
 2, Howard S. Wheater1, and Saman Razavi1

[revised manuscript text omitted]
 betweenof change over the cycles. There are some smallSmall oscillations are            |
| 616 | observed, indicating someminor numerical issues instabilities in the model, but they these do not cause     |
| 617 | major differences for the simulations. ForIn some cases/layers, the temperature keeps drifting (mostly      |
| 618 | cooling) for several hundred cycles before stabilizing (if itstabilization occurs). We note a few important |
| 619 | thingsfindings:                                                                                             |
|     |                                                                                                             |

[revised manuscript text omitted]

fully organic configuration (both ORG) is and M-org simulations are showing more variability in DZAA than the mineral one (M-org) but both match the depth deduced from observations for 01TC02- and both underestimate it. In general, matching ZODDZAA to observations is not an objective in itself but its occurrence well within the selected soil depth is more important. The largest value simulated is about 23m19m for HPC, which is less than half the total soil depth. ThatThis indicates that a smaller soil column

717 depth would not be recommendedsuitable for HPC but could be used for JMR and BWC.

**718**

**Possible Position of Figure 10**

719 Comparing to the observed envelopestemperature profiles for a selected year at each site (Figure 10), the 720 simulations look satisfactory in general.) reveals large difference between ORG and M-org configurations, 721 especially at HPC and BWC. The overall shapes of the profiles are captured depend on the selected 722 configuration. M-org works better for HPC while ORG is better at BWC. Both configurations do relatively 723 well for JMR and HPC despite the general over estimation of ALD for both sites.although this site is 724 characterized with deep peat. At BWC, the active layer depth-ORG simulation agrees well with 725 observations in terms of ALT but the temperature envelopes are generally colder than observed and gets 726 the. The M-org configuration at this site results in a talik between 2 and 9m which is not seen in the 727 observations. The minimum envelope getsis too cold near the surface for ORG configurations at the three 728 sites because of the thermal properties of the peat (Dobinski, 2011; Kujala et al., 2008). This is discussed 729 further in Section 3.5.

To aid with the selection of the best configuration for each site, we calculated RMSE for the temperature
 envelopes (Tmax and Tmin separately) by interpolating the simulation results at the depths of
 observations, discarding points/years where/when the sensors fail. The available records vary from site

[revised manuscript text omitted]